# AN EXPLANATION OF IN-CONTEXT LEARNING AS IMPLICIT BAYESIAN INFERENCE

**Sang Michael Xie, Aditi Raghunathan, Percy Liang, Tengyu Ma**
Stanford University
`{xie,aditir,pliang,tengyuma}@cs.stanford.edu`

## ABSTRACT

Large language models (LMs) such as GPT-3 have the surprising ability to do in-context learning, where the model learns to do a downstream task simply by conditioning on a prompt consisting of input-output examples. The LM learns from these examples *without being explicitly pretrained to learn*. Thus, it is unclear what enables in-context learning. In this paper, we study how in-context learning can emerge when pretraining documents have long-range coherence. Here, the LM must infer a latent document-level concept to generate coherent next tokens during pretraining. At test time, in-context learning occurs when the LM also infers a shared latent concept between examples in a prompt. We prove when this occurs despite a distribution mismatch between prompts and pretraining data in a setting where the pretraining distribution is a mixture of HMMs. In contrast to messy large-scale datasets used to train LMs capable of in-context learning, we generate a small-scale synthetic dataset (GINC) where Transformers and LSTMs both exhibit in-context learning[1]. Beyond the theory, experiments on GINC exhibit large-scale real-world phenomena including improved in-context performance with model scaling (despite the same pretraining loss), sensitivity to example order, and instances where zero-shot is better than few-shot in-context learning.

## 1 INTRODUCTION

Large language models (LMs) such as GPT-3 (Brown et al., 2020; Lieber et al., 2021; Wang & Komatsuzaki, 2021; Radford et al., 2019) are pretrained on massive text corpora to predict the next word given previous words. They demonstrate the surprising ability to do *in-context learning*, where an LM "learns" to do a task simply by conditioning on a prompt containing input-output pairs, achieving SOTA results on LAMBADA (Paperno et al., 2016) and TriviaQA (Joshi et al., 2017) tasks (18% and 3% over previous SOTA (Brown et al., 2020)). For example, consider the task of predicting nationalities from names. A prompt (Figure 1) is constructed by concatenating independent "training" examples (e.g., "Albert Einstein was German") followed by a "test example" ("Marie Curie was"). Conditioning on this prompt, GPT-3 places the largest probability on the correct output

$$p(\text{"Polish"} \mid \text{"Albert Einstein was German \textbackslash n Mahatma Gandhi was Indian \textbackslash n Marie Curie was"})$$

by inferring the task from examples. Intriguingly, GPT-3 was not explicitly pretrained to learn from examples, and the distribution of prompts (which concatenate independent examples) is quite different from natural language. Our understanding of in-context learning is limited since (i) real pretraining data is messy and (ii) in-context learning has so far required large-scale datasets and models.

In this paper, we introduce a simple pretraining distribution where in-context learning emerges. To generate a document, we first draw a latent concept $\theta$, which parameterizes the transitions of a Hidden Markov Model (HMM) (Baum & Petrie, 1966), then sample a sequence of tokens from the HMM (Figure 9). This latent variable structure is common in topic models such as LDA (Blei et al., 2003; Gruber et al., 2007). During pretraining, the LM must infer the latent concept across multiple sentences to generate coherent continuations. When conditioning on a prompt, in-context learning occurs when the LM also infers a shared *prompt concept* across examples to make a prediction. We assume the LM fits the pretraining distribution $p$ exactly with enough data and expressivity, so that the question of in-context learning becomes characterizing the conditional distribution of completions

---

[1]The code, data, and experiments are located on GitHub and CodaLab.

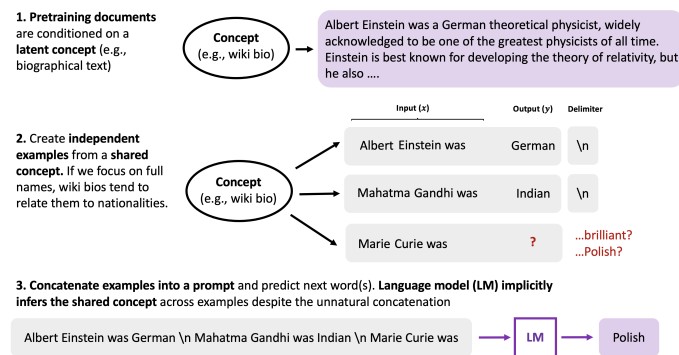

Figure 1: In-context learning can emerge from modeling long-range coherence in the pretraining data. During pretraining, the language model (LM) implicitly learns to infer a latent concept (e.g., wiki bios, which typically transition between name (Albert Einstein) → nationality (German) → occupation (physicist) → ...) shared across sentences in a document. Although prompts are unnatural sequences that concatenate independent examples, in-context learning occurs if the LM can still infer the shared concept across examples to do the task (name → nationality, which is part of wiki bios).

given prompts $p(\text{output}|\text{prompt})$ under the pretraining distribution, where the prompt is generated from a different distribution $p_{\text{prompt}}$. This conditional distribution, which is the *posterior predictive distribution*, marginalizes out the latent concepts:

$$p(\text{output}|\text{prompt}) = \int_{\text{concept}} p(\text{output}|\text{concept},\text{prompt})p(\text{concept}|\text{prompt})d(\text{concept}). \qquad (1)$$

If $p(\text{concept}|\text{prompt})$ concentrates on the prompt concept with more examples, then the LM learns via marginalization by "selecting" the prompt concept. Thus, in-context learning can be viewed as the LM implicitly performing Bayesian inference.

The main challenge is that prompts are sampled from a different distribution than the pretraining distribution. The canonical Bayesian asymptotic tool is the Bernstein-von Mises theorem (van der Vaart, 1998; Kleijn & van der Vaart, 2012; Gunst & Shcherbakova, 2008), which asserts (under regularity conditions) that the posterior distribution of a latent variable concentrates on the maximum likelihood estimate. However, Bernstein-von Mises typically assumes observations are independent and/or drawn from the same distribution as the model, both of which are not satisfied. We prove that despite the distribution mismatch, the asymptotic prediction error of in-context learning is optimal when the signal about the latent concept in each prompt example is larger than the error due to the distribution mismatch. Additionally, we prove that the in-context learning error decreases with the length of each example—thus, information in the inputs, not just the input-output mapping, can be useful for in-context learning.

As a companion to this theory, we created the **G**enerative **IN**-**C**ontext learning dataset (GINC), which is a small-scale synthetic dataset for studying in-context learning. We find that both Transformers (Vaswani et al., 2017) and LSTMs (Hochreiter & Schmidhuber, 1997) trained on GINC exhibit in-context learning. We verify intuitions from the theory, showing that the accuracy of in-context learning improves with the number of examples and example length. Ablations of the GINC dataset show that the latent concept structure in the pretraining distribution is crucial to the emergence of in-context learning.

The experiments also bring up open questions which go beyond our theory, which only studies the pretraining distribution. We find that scaling up the number of model parameters steadily improves the in-context accuracy despite achieving the same pretraining loss, showing that larger models may improve in-context learning beyond increasing the capacity for memorizing the training data better. Previously observed in-context learning phenomena such as sensitivity to example ordering (Zhao et al., 2021) and the existence of settings where zero-shot is better than one/few-shot learning (Brown et al., 2020) are also mirrored in GINC.

## 2    IN-CONTEXT LEARNING SETTING

**Pretraining distribution.**    In our framework, a latent concept $\theta$ from a family of concepts $\Theta$ defines a distribution over observed tokens $o$ from a vocabulary $\mathcal{O}$. To generate a document, we first sample a

concept from a prior $p(\theta)$ and then sample the document given the concept. Each pretraining document is a length $T$ sequence:

$$p(o_1,...,o_T) = \int_{\theta \in \Theta} p(o_1,...,o_T|\theta)p(\theta)d\theta. \tag{2}$$

We assume $p(o_1,...,o_T|\theta)$ is defined by a Hidden Markov Model (HMM). The concept $\theta$ determines the transition probability matrix of the HMM hidden states $h_1,...,h_T$ from a hidden state set $\mathcal{H}$.

**Prompt distribution.** The prompt distribution $p_{\text{prompt}}$ generates prompts for in-context learning. A prompt is a concatenation of $n$ independent training examples and 1 test input $x_{\text{test}}$, which are all conditioned on a shared prompt concept $\theta^*$. The goal is to predict the test output $y_{\text{test}}$ by predicting the next token conditioned on the prompt.

A prompt example is composed of an input token sequence $x$ (e.g., Albert Einstein was) followed by an output token $y$ (e.g., German). In particular, the $i$-th training example $O_i$ consists of an input $x_i = O_i[1:k-1]$ (the first $k-1$ tokens) followed by an output token $y_i = O_i[k]$ at the end[2]. The $i$-th training example is independently generated as follows:

1. Generate a start hidden state $h_i^{\text{start}}$ from a *prompt start distribution* $p_{\text{prompt}}$.
2. Given $h_i^{\text{start}}$, generate the example sequence $O_i = [x_i, y_i]$ from $p(O_i|h_i^{\text{start}}, \theta^*)$, the *pretraining distribution* conditioned on a prompt concept $\theta^*$.

The test input $x_{\text{test}} = x_{n+1}$ is sampled similarly. Between each example, there is a special delimiter token $o^{\text{delim}}$. The prompt consists of a sequence of training examples $(S_n)$ followed by the test example $x_{\text{test}}$:

$$[S_n, x_{\text{test}}] = [x_1, y_1, o^{\text{delim}}, x_2, y_2, o^{\text{delim}}, ..., x_n, y_n, o^{\text{delim}}, x_{\text{test}}] \sim p_{\text{prompt}}. \tag{3}$$

**Mismatch between prompt and pretraining distributions.** Since transitions between independent examples can be unnatural, the prompts are low probability sequences under the pretraining distribution. We provide a simple illustration using the names to nationalities example. Suppose that wiki bio documents in the pretraining data typically transition between name $\rightarrow$ nationality $\rightarrow$ occupation $\rightarrow$ .... In the prompt, the examples transition between name $\rightarrow$ nationality $\rightarrow$ name $\rightarrow$ nationality $\rightarrow$ ..., which contains low-probability transitions such as "German" $\rightarrow$ "Mahatma Gandhi". The prompt formatting (e.g., choice of delimiter) can also be a source of mismatch. We aim to show that despite this mismatch, large LMs can infer the prompt concept from examples.

**In-context predictor and task.** For in-context learning, the output target $y$ for each example $x$ is sampled according to $p_{\text{prompt}}(y|x)$:

$$y_{\text{test}} \sim p_{\text{prompt}}(y|x_{\text{test}}) = \mathbb{E}_{h_{\text{test}}^{\text{start}} \sim p_{\text{prompt}}(h_{\text{test}}^{\text{start}}|x_{\text{test}})} \left[ p(y|x_{\text{test}}, h_{\text{test}}^{\text{start}}, \theta^*) \right]. \tag{4}$$

where $h_{\text{test}}^{\text{start}}$ denotes the hidden state corresponding to the first token of $x_{\text{test}}$. We analyze the in-context predictor $f_n(x_{\text{test}}) = \arg\max_y p(y|S_n, x_{\text{test}})$, which outputs the most likely prediction over the *pretraining* distribution conditioned on the prompt from the *prompt* distribution[3]. We study the in-context predictor and its expected 0-1 error with $n$ examples $L_{0\text{-}1}(f_n) = \mathbb{E}_{x_{\text{test}}, y_{\text{test}} \sim p_{\text{prompt}}}[\mathbf{1}[f_n(x_{\text{test}}) \neq y_{\text{test}}]]$.

## 2.1 Assumptions

We detail the assumptions in our framework, including the structure of delimiters and regularity assumptions. We first assume that there exists a subset of *delimiter hidden states* $\mathcal{D}$ which generates the special delimiter token $o^{\text{delim}}$ deterministically.

**Assumption 1** (Delimiter hidden states). *Let the delimiter hidden states $\mathcal{D}$ be a subset of $\mathcal{H}$. For any $h^{delim} \in \mathcal{D}$ and $\theta \in \Theta$, $p(o^{delim}|h^{delim}, \theta) = 1$ and for any $h \notin \mathcal{D}$, $p(o^{delim}|h, \theta) = 0$.*

Thus, observing the delimiter $o^{\text{delim}}$ reveals that the corresponding hidden state is in $\mathcal{D}$, but does not reveal which element of $\mathcal{D}$ it is. The delimiter is usually a token that can appear in a broad range of contexts (e.g., newline). The delimiter ideally does not distract from the examples — for example, an adversarial delimiter could look like part of the input $x$. To mitigate these scenarios, we assume that no delimiter (e.g., newline) is significantly more likely under one concept rather than another.

---

[2]The example length $k$ is fixed for simplicity — we leave extending our analysis to variable $k$ as future work.
[3]In practice, greedy decoding or nucleus sampling (Holtzman et al., 2020) are used for likely completions.

**Assumption 2** (Bound on delimiter transitions). *For any delimiter state $h^{delim} \in \mathcal{D}$ and any hidden state $h \in \mathcal{H}$, the probability of transitioning to a delimiter hidden state under $\theta$ is upper bounded $p(h^{delim}|h,\theta) < c_2$ for any $\theta \in \Theta \setminus \{\theta^*\}$, and is lower bounded $p(h^{delim}|h,\theta^*) > c_1 > 0$ for $\theta^*$. Additionally, the start hidden state distribution for delimiter hidden states is bounded as $p(h^{delim}|\theta) \in [c_3,c_4]$.*

The prompt start distribution is a source of distribution shift that is separate from the shift from concatenating independent examples. We make an assumption that limits how much distribution shift is introduced by the prompt start distribution.

**Assumption 3** (Distribution shift from prompt start distribution). *We assume that the prompt start distribution $p_{prompt}$ is close in TV distance to all hidden transition distributions (under $\theta^*$) starting from a delimiter hidden state: $\max_{h^{delim} \in \mathcal{D}} TV(p_{prompt}(h) \| p(h|h^{delim},\theta^*)) < \Delta/4$. Here, $\Delta = p_{prompt}(y_{max}|x_{test}) - \max_{y \neq y_{max}} p_{prompt}(y|x_{test})$ is the margin between the most likely label $y_{max} = \mathrm{argmax}_y p_{prompt}(y|x_{test})$ and the second most likely label.*

Even if the maximum TV distance is 0, there is still distribution shift from concatenating independent examples. We also assume the prompt concept $\theta^*$ in the family $\Theta$, a broad set of concepts.

**Assumption 4** (Well-specification). *The prompt concept $\theta^*$ is in $\Theta$.*

Even though the pretraining distribution is broad, the prompt is still low probability under the pretraining distribution since it concatenates independent examples. Finally, if the prompt has zero probability under the prompt concept $\theta^*$, then Bayesian inference will not be able to infer the prompt concept as in Section 3.1. The following are regularity assumptions which mainly ensure that the prompt is not zero probability under $\theta^*$.

**Assumption 5** (Regularity). *The pretraining distribution $p$ satisfies: 1) Lower bound on transition probability for the prompt concept $\theta^*$: for any pair of hidden states $h,h' \in \mathcal{H}$, $p(h|h',\theta^*) > c_5 > 0$. 2) Start hidden state is lower bounded: for any $h \in \mathcal{H}$, $p(h|\theta^*) \geq c_8 > 0$. 3) All tokens can be emitted: for every symbol $o$, there is some hidden state $h \in \mathcal{H}$ such that $p(o|h,\theta^*) > c_6 > 0$, 4) The prior $p(\theta)$ has support over the entire concept family $\Theta$ and is bounded above everywhere.*

## 3 THEORETICAL ANALYSIS

We prove that in the limit of infinite examples, the error of the in-context predictor is optimal if a *distinguishability* condition holds — the prompt concept $\theta^*$ is distinct enough from the other concepts in $\Theta$ (e.g., when $\Theta$ is a discrete set). When distinguishability does not hold (e.g, $\Theta$ is continuous-valued), we show that the expected error still decreases with the length of each example, showing that information in both the inputs and the input-output mapping contribute to in-context learning.

### 3.1 HIGH-LEVEL APPROACH

Our goal is to show that $\mathrm{argmax}_y p(y|S_n,x_{test}) \to \mathrm{argmax}_y p_{prompt}(y|x_{test})$ as the number of examples $n$ grows. In the following, assume that the prompt has non-zero probability under the pretraining distribution $p$ given $\theta^*$, meaning that $p(S_n,x_{test}|\theta^*) > 0$. We expand $p(y|S_n,x_{test})$ to analyze its limit:

$$p(y|S_n,x_{test}) = \int_\theta p(y|S_n,x_{test},\theta)p(\theta|S_n,x_{test})d\theta$$

$$\propto \int_\theta p(y|S_n,x_{test},\theta)p(S_n,x_{test}|\theta)p(\theta)d\theta \quad \text{(Bayes' rule, drop the constant } \frac{1}{p(S_n,x_{test})})$$

$$= \int_\theta \sum_{h^{start}_{test} \in \mathcal{H}} p(y|x_{test},h^{start}_{test},\theta)p(h^{start}_{test}|S_n,x_{test},\theta)\frac{p(S_n,x_{test}|\theta)}{p(S_n,x_{test}|\theta^*)}p(\theta)d\theta \quad (5)$$

$$\text{(Law of total prob, Markov property, divide by } p(S_n,x_{test}|\theta^*) \text{ (a constant))}$$

$$= \int_\theta \sum_{h^{start}_{test} \in \mathcal{H}} p(y|x_{test},h^{start}_{test},\theta)p(h^{start}_{test}|S_n,x_{test},\theta)\exp(n \cdot r_n(\theta))p(\theta)d\theta \quad (6)$$

where $r_n(\theta) = \frac{1}{n}\log\frac{p(S_n,x_{test}|\theta)}{p(S_n,x_{test}|\theta^*)}$. In Theorem 1, we prove that under a distinguishability condition, $\exp(n \cdot r_n(\theta)) \to 0$ for all concepts $\theta$ except the prompt concept $\theta^*$, where $\exp(n \cdot r_n(\theta^*)) = 1$. The only nonzero term in the integral is when $\theta = \theta^*$, and thus the prompt concept is "selected" as a consequence of Bayesian inference[4]. Lemma 1 shows that the argmax after restricting to $\theta^*$ is the

---

[4]We can exchange limits and integrals since the probabilities are bounded (dominated convergence).

same as the most likely label under $p_{\text{prompt}}(y|x_{\text{test}})$ (using Assumption 3). Putting these together with Equation 6, the in-context predictor infers the prompt concept $\theta^*$:

$$\underset{y}{\arg\max}\, p(y|S_n,x_{\text{test}}) \to \underset{y}{\arg\max}\, p_{\text{prompt}}(y|x_{\text{test}}) \tag{7}$$

Thus, the in-context predictor is optimal as the number of in-context examples increases.

## 3.2 HEURISTIC DERIVATION

Recall from Section 3.1 that if $\exp(n \cdot r_n(\theta)) \to 0$ for all $\theta \neq \theta^*$, then Bayesian inference "selects" the prompt concept through marginalization. To do this, we focus on showing that $r_n(\theta)$, the average log-likelihood ratio between $\theta$ and $\theta^*$, converges to a negative constant, and thus $nr_n$ goes to $-\infty$.

The main technical challenge is to handle the sequence-of-examples structure of the prompt, which makes all the examples dependent with respect to the pretraining distribution. Our approach uses properties of delimiter tokens to approximately factorize the examples, with constant error per example. We let $O_i^{\text{ex}} = [o_{i-1}^{\text{delim}}, O_i]$ be the $i$-th input-output pair and the previous delimiter together for $i > 1$ and define $O_1^{\text{ex}} = O_1$. Expanding the likelihood term inside $r_n(\theta)$, our goal is to show

$$p(S_n, x_{\text{test}}|\theta) = p(x_{\text{test}}|S_n,\theta)p(S_n|\theta) \approx \prod_{i=1}^{n} O(1)p(O_i|\theta) \tag{8}$$

To show this, we expand $p(S_n|\theta)$ with the chain rule, and with Assumption 5 (to bound $p(x_{\text{test}}|S_n,\theta)$ by $O(1)$) it can be shown that

$$p(x_{\text{test}}|S_n,\theta)p(S_n|\theta) \approx \prod_{i=1}^{n} O(1)p(O_i^{\text{ex}}|O_{1:i-1}^{\text{ex}},\theta). \tag{9}$$

We then marginalize $p(O_i^{\text{ex}}|O_{1:i-1}^{\text{ex}},\theta)$ over the hidden state $h_{i-1}^{\text{delim}}$ corresponding to the delimiter in $O_i^{\text{ex}} = [o_{i-1}^{\text{delim}}, O_i]$:

$$\prod_{i=1}^{n} O(1)p(O_i^{\text{ex}}|O_{1:i-1}^{\text{ex}},\theta) = \prod_{i=1}^{n} O(1) \sum_{h_{i-1}^{\text{delim}} \in \mathcal{D}} p(O_i|h_{i-1}^{\text{delim}},\theta)p(h_{i-1}^{\text{delim}}|O_{1:i-1}^{\text{ex}},\theta) \approx \prod_{i=1}^{n} O(1)p(O_i|\theta) \tag{10}$$

While summing over $\mathcal{H}$ above would be a trivial equality, we can replace $\mathcal{H}$ with the set of delimiter hidden states $\mathcal{D}$ since $p(h|O_{1:i-1}^{\text{ex}},\theta) = 0$ for non-delimiter hidden states $h \notin \mathcal{D}$ (Assumption 1). We used in the first equality that $O_{1:i-1}^{\text{ex}} \to h_{i-1}^{\text{delim}} \to O_i^{\text{ex}}$ forms a Markov chain and $p(o_{i-1}^{\text{delim}}|h_{i-1}^{\text{delim}}) = 1$ (Assumption 1) to change $O_i^{\text{ex}}$ to $O_i$. Finally, we can show using properties of delimiter hidden states (Assumption 2) that $p(h_{i-1}^{\text{delim}}|O_{1:i-1}^{\text{ex}},\theta) = O(1)$ and $\sum_{h_{i-1}^{\text{delim}} \in \mathcal{D}} p(O_i|h_{i-1}^{\text{delim}},\theta) \approx O(1)p(O_i|\theta)$ in the second step. Therefore, we can upper bound $r_n(\theta)$ as

$$r_n(\theta) \leq \frac{1}{n}\left( O(n) + \sum_{i=1}^{n} \log\frac{p(O_i|\theta)}{p(O_i|\theta^*)} \right) \to O(1) + \mathbb{E}_{O \sim p_{\text{prompt}}}\left[ \log\frac{p(O|\theta)}{p(O|\theta^*)} \right]. \tag{11}$$

The expectation term can be written as the difference of two KL divergences, $KL(p_{\text{prompt}}(O)\|p(O|\theta^*)) - KL(p_{\text{prompt}}(O)\|p(O|\theta))$. We bound the first KL term by a constant using Assumption 5 — intuitively for one example, $p_{\text{prompt}}$ and $p(\cdot|\theta^*)$ are close. We break the second term into a sum of negative KL divergences over $k$ tokens. There are $O(k)$ KL terms and only $O(1)$ other error terms, which come from the distribution mismatch between the prompt and pretraining distributions. If the KL terms are larger than the error terms, then $r_n(\theta)$ has a negative limit. If this holds for all $\theta \neq \theta^*$, then we have $\exp(n \cdot r_n(\theta)) \to 0$ for all $\theta \neq \theta^*$, enabling in-context learning.

## 3.3 FORMAL RESULTS

### 3.3.1 IN-CONTEXT LEARNING UNDER DISTINGUISHABILITY

We define a distinguishability condition which formalizes when in-context learning occurs. Letting $p_\theta^j(o) \coloneqq p(O[j] = o|O[1:j-1],\theta)$ be the output distribution of the $j$-th token given the previous tokens and $p_{\text{prompt}}^j(o) \coloneqq p_{\text{prompt}}(O[j] = o|O[1:j-1])$ be the analogous distribution under the prompt

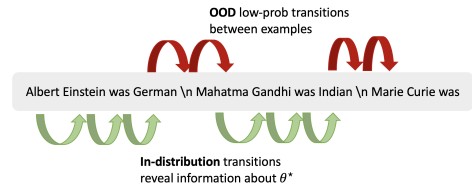

Figure 2: When the signal about the prompt concept within each example (green) is greater than the error from low-probability transitions between examples, in-context learning succeeds in our latent concept setting (Theorem 1). Increasing the example length $k$ increases the signal. The signal for in-context learning comes from tokens in both the inputs and the input-output mapping.

distribution, the distinguishability condition depends on the KL divergence between $p_{\text{prompt}}^j$ (which represents $\theta^*$) and $p_\theta^j$ as well as error terms $\epsilon_{\text{start}}^\theta$ and $\epsilon_{\text{delim}}^\theta$ coming from the distribution mismatch between the prompt and pretraining distributions at the start and delimiter token for each example:

$$KL_j(\theta^*\|\theta) := \mathbb{E}_{O[1:j-1]\sim p_{\text{prompt}}}[KL(p_{\text{prompt}}^j\|p_\theta^j)] \tag{12}$$

$$\epsilon_{\text{delim}}^\theta := 2(\log(c_2) - \log(c_1)) + \log(c_4) - \log(c_3), \quad \epsilon_{\text{start}}^\theta := \log(1/c_8). \tag{13}$$

**Condition 1** (Distinguishability). *We define $\theta^*$ to be distinguishable if for all $\theta \in \Theta, \theta \neq \theta^*$,*

$$\sum_{j=2}^k KL_j(\theta^*\|\theta) > \epsilon_{start}^\theta + \epsilon_{delim}^\theta. \tag{14}$$

When the signal from KL divergence (LHS) is larger than the error terms, Equation 14 is satisfied (Figure 2). For larger example lengths $k$, the LHS increases, improving distinguishability. Intuitively, larger example lengths increase the proportion of the prompt sampled from the pretraining distribution by providing more evidence for Bayesian inference. Under Condition 1, the in-context predictor asymptotically achieves the optimal expected error.

**Theorem 1.** *Assume the assumptions in Section 2.1 hold. If Condition 1 holds, then as $n \to \infty$ the prediction according to the pretraining distribution is*

$$\operatorname*{argmax}_y p(y|S_n, x_{test}) \to \operatorname*{argmax}_y p_{prompt}(y|x_{test}). \tag{15}$$

*Thus, the in-context predictor $f_n$ achieves the optimal 0-1 risk: $\lim_{n\to\infty} L_{\text{0-1}}(f_n) = \inf_f L_{\text{0-1}}(f)$.*

### 3.3.2 NON-DISTINGUISHABLE CASE

The distinguishability condition (Condition 1) fails when there is some $\theta \neq \theta^*$ for which the KL divergence between $\theta$ and $\theta^*$ is less than the error terms. However, this also means that the output distributions of $\theta$ and $\theta^*$ are close in KL. We leverage this to prove that the expected 0-1 error decreases with the example length $k$ under two different settings where distinguishability does not hold.

**Continuity.** Our first result relies on a continuity assumption between the concept parameter and its corresponding output distribution. Our assumption is based on prior works (Kleijn & van der Vaart, 2012), where the KL divergence is assumed to have a 2nd-order Taylor expansion.

**Theorem 2.** *Let the set of $\theta$ which does not satisfy Equation 14 in Condition 1 to be $\mathcal{B}$. Assume that KL divergences have a 2nd-order Taylor expansion around $\theta^*$:*

$$\forall j > 1, \ KL_j(\theta^*\|\theta) = \frac{1}{2}(\theta - \theta^*)^\top I_{j,\theta^*}(\theta - \theta^*) + O(\|\theta - \theta^*\|^3) \tag{16}$$

*where $I_{j,\theta^*}$ is the Fisher information matrix of the $j$-th token distribution with respect to $\theta^*$. Let $\gamma_{\theta^*} = \frac{\max_j \lambda_{max}(I_{j,\theta^*})}{\min_j \lambda_{min}(I_{j,\theta^*})}$ where $\lambda_{max}, \lambda_{min}$ return the largest and smallest eigenvalues. Then for $k > 1$ and as $n \to \infty$, the 0-1 risk of the in-context learning predictor $f_n$ is bounded as*

$$\lim_{n\to\infty} L_{\text{0-1}}(f_n) \leq \inf_f L_{\text{0-1}}(f) + g^{-1}\left(O\left(\frac{\gamma_{\theta^*}\sup_{\theta\in\mathcal{B}}(\epsilon_{start}^\theta + \epsilon_{delim}^\theta)}{k-1}\right)\right) \tag{17}$$

*where $g(\delta) = \frac{1}{2}((1-\delta)\log(1-\delta) + (1+\delta)\log(1+\delta))$ is a calibration function (Steinwart, 2007; Ávila Pires & Szepesvári, 2016) for the multiclass logistic loss for $\delta \in [0,1]$.*

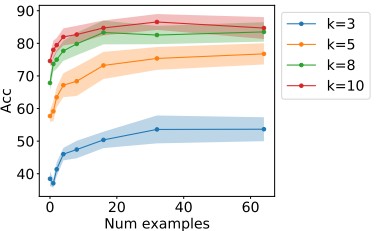 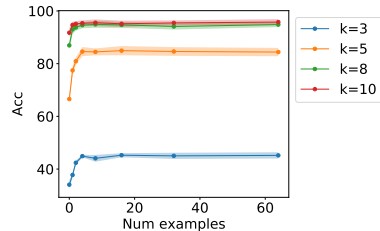

Figure 3: In-context accuracy (95% intervals) of Transformers (left) and LSTMs (right) on the GINC dataset. Accuracy increases with number of examples $n$ and length of each example $k$.

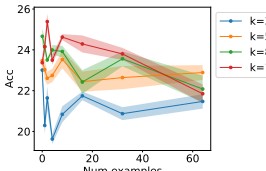 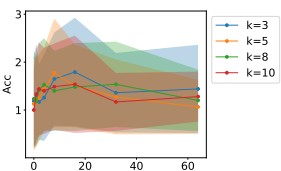 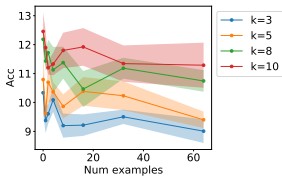

Figure 4: Ablation studies for 4 layer Transformers on the GINC dataset with vocab size 50. **(Left)** When pretrained with only one concept, in-context learning fails. **(Middle)** When the pretraining data has random transitions, the model sees all token transitions but in-context learning fails. **(Right)** When prompts are from random unseen concepts, in-context learning fails to extrapolate.

Since the inverse calibration function $g^{-1}$ is roughly linear in $\epsilon$ for $\epsilon \leq 0.7$, the excess risk roughly decreases as $O(1/k)$. When the "worst-case condition number" $\gamma_{\theta^*}$ of the Fisher information matrices is smaller (well-conditioned), the error decreases. Intuitively, this means that there is no direction to vary $\theta^*$ in which the output distribution will sharply change. As a consequence, the concepts $\theta$ that are not distinguishable from the prompt concept $\theta^*$ parameterize distributions that produce similar outputs to the prompt concept and thus achieve a small error.

**Varying-length test examples.** In the setting where the length of $x_{\text{test}}$ is random (uniformly from 2 to $k$), we can give a similar error guarantee without continuity.

**Theorem 3.** *Let the set of $\theta$ which does not satisfy Equation 14 in Condition 1 to be $\mathcal{B}$. Let the length of the test example $x_{test}$ be uniformly distributed between 2 and $k$, for $k \geq 2$. Then for $k \geq 2$ and as $n \to \infty$, the 0-1 risk of the in-context learning predictor $f_n$ is bounded as*

$$\lim_{n \to \infty} L_{\text{0-1}}(f_n) \leq \inf_f L_{\text{0-1}}(f) + g^{-1}\left( O\left( \frac{\sup_{\theta \in \mathcal{B}}(\epsilon_{start}^\theta + \epsilon_{delim}^\theta)}{k-1} \right) \right). \tag{18}$$

Instead of measuring only the error at the $k$-th token, we average the prediction error on the 2nd to $k$-th tokens. However, we leave bridging the mismatch between training examples, which are consistently length $k$, and test examples, which have random length, to future work.

## 4 SIMULATIONS

We generate the GINC dataset and show that Transformers (Vaswani et al., 2017) and LSTMs (Hochreiter & Schmidhuber, 1997) trained on GINC exhibit in-context learning. In the theory, we assumed that the pretrained LM fits the pretraining distribution exactly. Here, we pretrain LMs to approximate the pretraining distribution and find that the in-context learning properties transfer to the LM.

**GINC dataset.** We construct the GINC dataset according to our theory (see Appendix F.1). For pretraining, we define a uniform mixture of HMMs over a family $\Theta$ of 5 concepts to generate 1000 pretraining documents with ~10 million tokens total. For prompting, we generate prompts with 0 to 64 training examples and example lengths $k \in \{3,5,8,10\}$ (2500 prompts for each setting). The target token $y_{\text{test}}$ is taken to be the most likely output instead of sampling so that the intrinsic error is 0.

**Main result.** We train GPT-2-based Transformers (Radford et al., 2019) and LSTMs on three versions of the GINC dataset with vocabulary sizes 50, 100, and 150, then evaluate the in-context

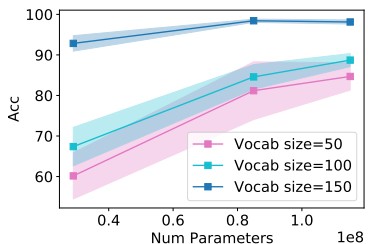

Figure 5: In-context accuracy (95% intervals) of Transformers improves as model size increases on the GINC dataset for vocabulary sizes 50, 100, and 150.

| Model | # Params | Train loss (pretraining) | Val loss (pretraining) | In-context Acc |
|---|---|---|---|---|
| Vocab size 50, $k=10, n=64$ | | | | |
| Transformer (4 layer) | 29M | 1.49 | 1.50 | $60.2 \pm 5.7$ |
| Transformer (12 layer) | 85M | 1.31 | 1.33 | $81.2 \pm 7.1$ |
| Transformer (16 layer) | 115M | 1.31 | 1.33 | $84.7 \pm 3.4$ |
| LSTM | 28M | 1.31 | 1.35 | $95.8 \pm 1.11$ |
| Vocab size 100, $k=10, n=64$ | | | | |
| Transformer (4 layer) | 29M | 1.58 | 1.59 | $67.4 \pm 4.7$ |
| Transformer (12 layer) | 85M | 1.40 | 1.42 | $84.6 \pm 3.0$ |
| Transformer (16 layer) | 115M | 1.41 | 1.43 | $88.7 \pm 1.6$ |
| LSTM | 28M | 1.43 | 1.44 | $95.8 \pm 1.54$ |
| Vocab size 150, $k=10, n=64$ | | | | |
| Transformer (4 layer) | 29M | 1.44 | 1.45 | $92.8 \pm 1.9$ |
| Transformer (12 layer) | 85M | 1.27 | 1.28 | $98.4 \pm 0.4$ |
| Transformer (16 layer) | 115M | 1.27 | 1.28 | $98.1 \pm 0.5$ |
| LSTM | 28M | 1.26 | 1.31 | $99.2 \pm 1.06$ |

Figure 6: In-context accuracies (95% intervals) on GINC with vocab sizes (50, 100, 150) for Transformers and LSTMs. Accuracy improves with scale even though the pretraining loss may be the same.

accuracy (see Appendix F.2, F.3). We average all results over 5 pretraining runs. Figure 3 shows that for both Transformer and LSTMs, in-context accuracy improves as the number of prompt examples $n$ and the example length $k$ increase, verifying our theory.

**Ablations on the latent concept structure.** We ablate the role of the mixture-of-concepts structure in GINC. In Figure 4 (left), we pretrain a 4 layer Transformer on data with only one concept (removing the prior) from $\Theta$, resulting in flat in-context learning curves. Figure 4 (middle) shows that pretraining on random pretraining data, which contains all possible token transitions, in-context learning also fails. Therefore, the mixture-of-concepts structure is important and simply seeing diverse token transitions does not enable in-context learning.

**Extrapolation to unseen concepts.** Full generative control of GINC allows for experimentation with latent variables in the pretraining distribution. For example, in large-scale datasets, it is difficult to test whether a concept or task is in the pretraining data. We test this in GINC by testing the in-context accuracy of a 4 layer Transformer on prompts generated from 5 random concepts that are not in the pretraining family of concepts. Figure 4 (right) shows that in-context learning also fails for these novel concepts.

**Effect of model size and architecture.** Figure 5 shows that increasing the size of the Transformer (4, 12, 16 layers) steadily increases the in-context accuracy, corroborating the results of Brown et al. (2020). Table 6 shows that even though larger Transformers may have the same pretraining loss (e.g., 12 and 16 layer Transformers both get 1.33 validation loss for vocab size 50), the in-context accuracy still improves (81% to 85% from 12 to 16 layers), suggesting that larger models can improve in-context learning beyond improving pretraining perplexity. This may be related to phenomena from overparameterization and overtraining (Zhang et al., 2017; Power et al., 2021). Finally, the model architecture also plays a role — LSTMs consistently outperform Transformers on GINC despite having fewer parameters, perhaps due to the similarity between HMMs and LSTMs. We leave analysis of the effect of model scaling and model architecture as open questions.

**Sensitivity to example ordering.** In Figure 7 (left), we test the sensitivity of in-context accuracy on GINC to the ordering of the prompt examples, following Zhao et al. (2021). For this experiment, we consider prompts generated from a single concept and prompt start distribution. We sample 10 different sets (leading to 10 training set IDs) of 4 examples and generate all 24 possible permutations for each example set. We consider the in-context accuracy of the 4 layer Transformer trained on GINC with vocabulary size 50. Similarly to the behavior of GPT-3 (Zhao et al., 2021), there is a significant variation (10–40% difference) between permutations of the same set of examples.

**Zero-shot is sometimes better than few-shot.** In some settings in GINC, we find that zero-shot performance can be better than few-shot performance. This mirrors GPT-3 on some datasets (e.g., LAM-BADA, HellaSwag, PhysicalQA, RACE-m, CoQA/SAT analogies for smaller models (Brown et al., 2020)). This occurs especially when the transition probabilities in GINC are lower entropy (controlled via a temperature parameter). For this experiment, we consider GINC with transition matrix temperature parameter 0.01 (instead of 0.1), 12 concepts, and vocabulary size 100. Figure 7 (right) shows that here, few-shot accuracy is initially worse than zero-shot accuracy, but can recover with more examples. We hypothesize that the distracting prompt structure initially decreases the accuracy in this setting.

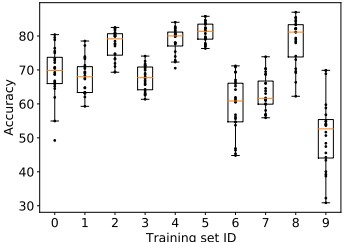 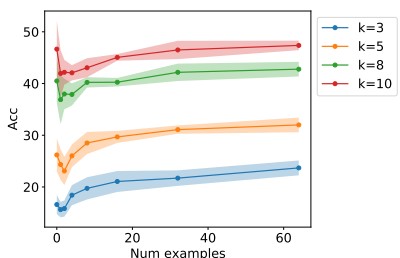

Figure 7: **(Left)** In-context accuracy varies widely with example ordering. Each training ID refers to a set of training examples. Each dot refers to the in-context learning accuracy of one permutation of the training examples for that particular training ID. **(Right)** Zero-shot performance can be higher than one/few-shot performance in some settings in GINC, mirroring the behavior of GPT-3 on some datasets such as LAMBADA (Brown et al., 2020). The few-shot setting introduces the distracting prompt structure, which can initially lower accuracy.

## 5 DISCUSSION AND RELATED WORK

**Learning via Bayesian inference and extrapolation.**    The canonical Bernstein-von Mises theorem (van der Vaart, 1998) does not apply for in-context learning since the prompt examples are not independent under the pretraining distribution. Gunst & Shcherbakova (2008) show a Bernstein-von Mises-type result for observations from an HMM, but do not handle observations from a different distribution. Future directions include more precise asymptotic results about the posterior distribution and results under misspecification/extrapolation (Kleijn & van der Vaart, 2012). A possible avenue for extrapolation to some types of unseen concepts is to factorize the latent concept into semantics and syntax. While the pretraining data may contain only some semantics-syntax pairs, the language model could generalize to unseen pairs if it learns generalizable syntactical operations such as copying or reordering.

**Topic models and HMMs.**    Topic models such as LDA (Blei et al., 2003) also have document-level latent variables, but learning is typically relies on algorithms such as EM (Dempster et al., 1977), variational inference (Jordan et al., 1999), or MCMC (Metropolis et al., 1953; Hastings, 1970). We focus on learning as a natural result of Bayesian inference without an explicit inference algorithm. Wei et al. (2021a) also use an HMM model in their pretraining analysis. However, they analyze how pre-trained representations learned with masked LMs (Devlin et al., 2019; Liu et al., 2019; Lewis et al., 2020; Clark et al., 2020) can improve optimization-based downstream learning (Li & Liang, 2021; Lester et al., 2021) rather than in-context learning.

**Bridging the mismatch between pretraining and prompting.**    Prior works support our theoretical intuitions that reducing the prompt distribution mismatch would improve in-context learning. Finetuning LMs on text with a prompting format improves its zero-shot performance (Wei et al., 2021b; Sanh et al., 2021) and optimizing prompt templates improves few-shot finetuning (Jiang et al., 2020; Schick & Schütze, 2021; Shin et al., 2020; Gao et al., 2021). Zhao et al. (2021); Holtzman et al. (2021) improve in-context accuracy via calibration or renormalization, a form of adaptation to the prompt distribution.

**Meta-learning.**    Meta-learning methods can also train a sequence model to learn from examples (Ravi & Larochelle, 2017). However, meta-learning models are trained to learn, while in-context learning emerges from LM pretraining.

**Studying large-scale phenomena at a small scale.**    We can study in-context learning, a large scale phenomenon, at a small scale in GINC because the complexity of the pretraining distribution (HMM hidden state size, number of latent concepts) is small, such that the data and models are relatively larger. Since GINC is synthetic, we can also control the latent data properties (e.g., unseen concepts) to make predictions about large LMs while working at a small scale.

## 6 CONCLUSION

We cast in-context learning as implicit Bayesian inference, where the pretrained LM implicitly infers a concept when making a prediction. We show that in-context learning occurs when the pre-training distribution is a mixture of HMMs. Our work provides a first step towards understanding in-context learning, which we hope will provide insight for improving pretraining and prompting.

ACKNOWLEDGEMENTS

We thank Tianyi Zhang, Frieda Rong, Lisa Li, Colin Wei, Shibani Santurkar, Tri Dao, Ananya Kumar, and Shivam Garg for helpful discussions and feedback. SMX is supported by an NDSEG Fellowship. The work is partially supported by an Open Philanthropy Project Award, SDSI, and SAIL at Stanford University. TM acknowledges support of Google Faculty Award, NSF IIS 2045685, the Sloan Fellowship, and JD.com. Toyota Research Institute provided funds to support this work.

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

## A   FRAMEWORK DETAILS

**Prompt distribution details.**   For in-context learning, we sample a prompt from a new distribution $p_{\text{prompt}}$, which consists of $n$ independent training examples and 1 test example. We first sample $n$ hidden segments $H$ of length $k$ by sampling the first element $h^{\text{start}} = H[1]$ from a prompt start distribution $p_{\text{prompt}}$. Then, we sample the rest of the segment $H^{\text{seg}} = H[2:k]$ from the hidden transition distribution of the pretraining distribution $p$ corresponding to a particular concept $\theta^*$:

$$H_1,...,H_n, \quad H_i = [h_{i,1},...,h_{i,k}] \tag{19}$$

$$h_i^{\text{start}} = H_i[1] \sim p_{\text{prompt}}, \quad H_i^{\text{seg}} = H_i[2:k] \sim p(H_i^{\text{seg}}|h^{\text{start}},\theta^*). \tag{20}$$

To end each example (except the test example), we sample $n$ delimiters $h^{\text{delim}} \in \mathcal{D}$ from $p_{\text{prompt}}^{\text{delim}}$:

$$h_1^{\text{delim}},...,h_n^{\text{delim}}, \quad h_i^{\text{delim}} \sim p_{\text{prompt}}^{\text{delim}}. \tag{21}$$

Conditioned on hidden variables $H_i$ and $h_i^{\text{delim}}$, we sample the observed tokens $O_i = [o_{i,1},...,o_{i,k}]$ and $o_i^{\text{delim}}$ respectively from the pre-training distribution:

$$O_1,...,O_n, \quad O_i \sim p(O_i|H_i) \tag{22}$$

$$o_1^{\text{delim}},...,o_n^{\text{delim}}, \quad o_i^{\text{delim}} \sim p(o_i^{\text{delim}}|h_i^{\text{delim}},\theta^*) \tag{23}$$

The "input" for each example is $x_i = O_i[1:k-1]$ and the "output" is $y_i = O_i[k]$. Taking $S$ to be the sequence of training examples (without the test example), the resulting prompt sequence is

$$[S_n,x_{\text{test}}] = [O_1,o_1^{\text{delim}},...,O_n,o_n^{\text{delim}},x_{\text{test}}] = [x_1,y_1,o_1^{\text{delim}},x_2,y_2,o_2^{\text{delim}},...,x_n,y_n,o_n^{\text{delim}},x_{\text{test}}] \sim p_{\text{prompt}} \tag{24}$$

where $x_{\text{test}} = x_{n+1} = O_{n+1}[1:k-1]$ is sampled via the same process but with $k-1$ elements.

## B   PROPOSITIONS FOR THEOREM 1

The following propositions, which lower bound the probability of a delimiter token and probability of an example under $\theta^*$, are direct corollaries of the assumptions.

**Proposition 1.**   *For all $i$, we have $p(h_i^{delim}|O_1,o_1^{delim},...,O_i,\theta^*) > c_1$ and $p(h_i^{delim}|O_1,o_1^{delim},...,O_i,\theta) < c_2$.*

*Proof.*   By Assumption 2,

$$p(h_i^{\text{delim}}|O_1,o_1^{\text{delim}},...,O_i,\theta) = \sum_{h_{i,k}} p(h_i^{\text{delim}}|h_{i,k})p(h_{i,k}|O_1,o_1^{\text{delim}},...,O_i,\theta) \tag{25}$$

$$< \sum_{h_{i,k}} c_2 p(h_{i,k}|O_1,o_1^{\text{delim}},...,O_i,\theta) = c_2. \tag{26}$$

Similarly,

$$p(h_i^{\text{delim}}|O_1,o_1^{\text{delim}},...,O_i,\theta^*) = \sum_{h_{i,k}} p(h_i^{\text{delim}}|h_{i,k})p(h_{i,k}|O_1,o_1^{\text{delim}},...,O_i,\theta^*) \tag{27}$$

$$> \sum_{h_{i,k}} c_1 p(h_{i,k}|O_1,o_1^{\text{delim}},...,O_i,\theta^*) = c_1. \tag{28}$$

$\square$

**Proposition 2.**   *The probability of an example is lower bounded for $\theta^*$: there is some $c_7 > 0$ such that $p(O_i|h_i^{start},h_{j,l},\theta^*) > c_7$ for all $i$ and future hidden states $h_{j,l}$, for any $l$ and $j > i$.*

*Proof.*   By Assumption 5, we have

$$p(O_i|h_i^{\text{start}},h_{j,l},\theta^*) = \sum_{H_i} p(O_i|H_i)p(H_i|h_i^{\text{start}},h_{j,l},\theta^*) > (c_6)^k \tag{29}$$

for some $H_i$. We have

$$p(H_i|h_i^{\text{start}},h_{j,l},\theta^*) = \frac{p(h_{j,l}|H,h_i^{\text{start}},\theta^*)p(H|h_i^{\text{start}},\theta^*)}{p(h_{j,l}|h_i^{\text{start}},\theta^*)} > c_5^2 \tag{30}$$

which lower bounds the terms in the numerator by $c_5$ (marginalizing over previous hidden states), and upper bounding the denominator by 1. Setting $c_7 = (c_6)^k c_5^2$ finishes the proof. $\square$

## C  CONVERGENCE OF THE IN-CONTEXT PREDICTOR

Under Assumption 3, we show that the in-context predictor $f_n(x_\text{test}) = \text{argmax}_y \, p(y|S_n, x_\text{test})$ converges when abstracting away the Bayesian inference component (the selection of $\theta^*$ from $\Theta$) of the in-context predictor. We will complete the argument for the convergence of the in-context predictor in the proof of Theorem 1.

**Lemma 1.** *Suppose the prompt $S_n$ and the test input $x_{test}$ are given. Under Assumption 3, we show that the argmax of the averaged predictive distribution conditioned on $\theta^*$ and a prompt $S_n$ is the same as the argmax of the prompt predictive distribution:*

$$\underset{y}{\text{argmax}} \sum_{h_{test}^{start} \in \mathcal{H}} p(y|x_{test}, h_{test}^{start}, \theta^*) p(h_{test}^{start}|S_n, x_{test}, \theta^*) = \underset{y}{\text{argmax}} \, p_{prompt}(y|x_{test}). \tag{31}$$

*Proof.* First, we note by definition that

$$p_\text{prompt}(y|x_\text{test}) = \sum_{h_\text{test}^\text{start} \in \mathcal{H}} p(y|x_\text{test}, h_\text{test}^\text{start}, \theta^*) p_\text{prompt}(h_\text{test}^\text{start}|x_\text{test}). \tag{32}$$

Expanding the last term, we have

$$p_\text{prompt}(h_\text{test}^\text{start}|x_\text{test}) \propto p(x_\text{test}|h_\text{test}^\text{start}, \theta^*) p_\text{prompt}(h_\text{test}^\text{start}). \tag{33}$$

which is proportional to a constant in $x_\text{test}$.

On the other hand, analyzing one term inside the LHS of the lemma statement, we have

$$p(h^\text{start}|S_n, x_\text{test}, \theta^*) \propto p(x_\text{test}|h_\text{test}^\text{start}, \theta^*) p(h_\text{test}^\text{start}|S_n, \theta^*) \tag{34}$$

which is proportional to a constant in $x_\text{test}$ and $S_n$. The quantities differ in the last term, which we expand below and put in matrix form. Let $T \in \mathbb{R}^{|\mathcal{H}| \times |\mathcal{D}|}$ be the matrix that represents the transition probabilities starting from a delimiter state: $p(h_\text{test}^\text{start}|h^\text{delim})$ for $h_\text{test}^\text{start} \in \mathcal{H}$ and $h^\text{delim} \in \mathcal{D}$. As a result,

$$p(h_\text{test}^\text{start}|S_n, \theta^*) = \sum_{h_n^\text{delim}} p(h_\text{test}^\text{start}|h_n^\text{delim}, \theta^*) p(h_n^\text{delim}|S_n, \theta^*) \tag{35}$$

$$= Tv \tag{36}$$

where $h_n^\text{delim}$ is the delimiter hidden state before $h_\text{test}^\text{start}$.

Let $W \in \mathbb{R}^{|\mathcal{Y}| \times |\mathcal{H}|}$ be the matrix that represents the probabilities $p(y|x_\text{test}, h_\text{test}^\text{start}, \theta^*) p(x_\text{test}|h_\text{test}^\text{start}, \theta^*)$ for all the possible $y \in \mathcal{Y}$ and $h_\text{test}^\text{start} \in \mathcal{H}$. Overall, we can write

$$\sum_{h_\text{test}^\text{start} \in \mathcal{H}} p(\cdot|x_\text{test}, h_\text{test}^\text{start}, \theta^*) p(h_\text{test}^\text{start}|S_n, x_\text{test}, \theta^*) = WTv \tag{37}$$

$$p_\text{prompt}(\cdot|x_\text{test}) = Wu \tag{38}$$

where $u \in \mathbb{R}^{|\mathcal{H}|}$ is the vector of probabilities that corresponds to the prompt start distribution $p_\text{prompt}$.

Bounding the difference between the two predictive distributions,

$$\|WTv - Wu\|_\infty \leq \|WTv - Wu\|_1 \tag{39}$$

$$= \sum_{i=1}^{|\mathcal{Y}|} |W_i^\top (Tv - u)|_i \tag{40}$$

$$= \sum_{i=1}^{|\mathcal{Y}|} \left| \sum_{j=1}^{|\mathcal{H}|} W_{ij}(Tv - u)_j \right| \tag{41}$$

$$\leq \sum_{i=1}^{|\mathcal{Y}|} \sum_{j=1}^{|\mathcal{H}|} W_{ij} |(Tv - u)_j| \quad (W_{ij} \geq 0) \tag{42}$$

$$= \sum_{j=1}^{|\mathcal{H}|} \left( \sum_{i=1}^{|\mathcal{Y}|} W_{ij} \right) |(Tv - u)_j| \tag{43}$$

$$= \|Tv - u\|_1. \tag{44}$$

Using Assumption 3, we can further bound this by $\Delta/2$:

$$\|Tv-u\|_1 = 2TV\left(p_{\text{prompt}}(\cdot)\|\sum_{i=1}^{|\mathcal{D}|} v_i p(\cdot|h^{\text{delim}}=i,\theta^*)\right) \tag{45}$$

$$\leq 2\sum_{i=1}^{|\mathcal{D}|} v_i TV(p_{\text{prompt}}(\cdot)\|p(\cdot|h^{\text{delim}}=i,\theta^*)) \quad \text{(convexity of TV distance)} \tag{46}$$

$$\leq 2\max_{h^{\text{delim}}\in\mathcal{D}} TV(p_{\text{prompt}}(\cdot)\|p(\cdot|h^{\text{delim}},\theta^*)) < \Delta/2. \tag{47}$$

Since the probability of any output does not change by more than $\Delta/2$ and the margin between the most likely label and the second most likely label is $\Delta$, the argmax's are the same, showing the result. $\quad\square$

## D  PROOF OF THEOREM 1

*Proof.* We analyze the most likely prediction over the pretraining distribution conditioned on the prompt $\text{argmax}_y p(y|S_n,x_{\text{test}})$.

$$p(y|S_n,x_{\text{test}}) = \int_\theta p(y|S_n,x_{\text{test}},\theta)p(\theta|S_n,x_{\text{test}})d\theta \tag{48}$$

$$\propto \int_\theta p(y|S_n,x_{\text{test}},\theta)p(S_n,x_{\text{test}}|\theta)p(\theta)d\theta \tag{49}$$

$$\propto \int_\theta p(y|S_n,x_{\text{test}},\theta)\frac{p(S_n,x_{\text{test}}|\theta)}{p(S_n,x_{\text{test}}|\theta^*)}p(\theta)d\theta \tag{50}$$

$$= \int_\theta \sum_{h_{\text{test}}^{\text{start}}\in\mathcal{H}} p(y|x_{\text{test}},h_{\text{test}}^{\text{start}},\theta)p(h_{\text{test}}^{\text{start}}|S_n,x_{\text{test}},\theta)\frac{p(S_n,x_{\text{test}}|\theta)}{p(S_n,x_{\text{test}}|\theta^*)}p(\theta)d\theta \tag{51}$$

Defining the following quantity,

$$r_n(\theta) = \frac{1}{n}\log\frac{p(S_n,x_{\text{test}}|\theta)}{p(S_n,x_{\text{test}}|\theta^*)}. \tag{52}$$

we will show that under distinguishability for all $\theta\neq\theta^*$, $r_n(\theta)$ converges to a negative constant such that

$$\frac{p(S_n,x_{\text{test}}|\theta)}{p(S_n,x_{\text{test}}|\theta^*)} = \exp(n\cdot r_n(\theta))\to 0 \tag{53}$$

for $\theta\neq\theta^*$, whereas this ratio is always 1 for $\theta=\theta^*$. This will then "select" the desired prompt concept through marginalization.

Supposing that Equation 53 holds, we show that the theorem statement holds. Let

$$\Delta' = \max_{h^{\text{delim}}\in\mathcal{D}} TV(p_{\text{prompt}}(\cdot)\|p(\cdot|h^{\text{delim}},\theta^*)) < \Delta/2, \tag{54}$$

and let $\epsilon < (\Delta/2-\Delta')p(\theta^*)$. Then for $n$ large enough (due to Equation 53),

$$\int_\theta \sum_{h_{\text{test}}^{\text{start}}\in\mathcal{H}} p(y|x_{\text{test}},h_{\text{test}}^{\text{start}},\theta)p(h_{\text{test}}^{\text{start}}|S_n,x_{\text{test}},\theta)\frac{p(S_n,x_{\text{test}}|\theta)}{p(S_n,x_{\text{test}}|\theta^*)}p(\theta)d\theta \tag{55}$$

$$= \sum_{h_{\text{test}}^{\text{start}}\in\mathcal{H}} p(y|x_{\text{test}},h_{\text{test}}^{\text{start}},\theta^*)p(h_{\text{test}}^{\text{start}}|S_n,x_{\text{test}},\theta^*)p(\theta^*) + \int_{\theta\neq\theta^*}\epsilon_\theta(y)p(\theta)d\theta \tag{56}$$

$$\propto \sum_{h_{\text{test}}^{\text{start}}\in\mathcal{H}} p(y|x_{\text{test}},h_{\text{test}}^{\text{start}},\theta^*)p(h_{\text{test}}^{\text{start}}|S_n,x_{\text{test}},\theta^*) + \frac{1}{p(\theta^*)}\int_{\theta\neq\theta^*}\epsilon_\theta(y)p(\theta)d\theta \tag{57}$$

where $\epsilon_\theta(y)\leq\epsilon/2$ for all $y\in\mathcal{Y}$.

By Lemma 1, the argmax of the first term of Equation 57 is the same as $\text{argmax}_y p_{\text{prompt}}(y|x_{\text{test}})$, where the margin between the most likely label and the second most likely is at least $\Delta/2-\Delta'$. Since

$$\frac{1}{p(\theta^*)}\int_{\theta\neq\theta^*}\epsilon_\theta(y)p(\theta) \leq \frac{\epsilon}{2p(\theta^*)} < (\Delta/2-\Delta')/2 \tag{58}$$

for all $y \in \mathcal{Y}$, the argmax of Equation 57 is also the same as $\operatorname{argmax} p_{\text{prompt}}(y|x_{\text{test}})$.

Now it remains to show that $r_n(\theta)$ converges to a negative constant for $\theta \neq \theta^*$. Let $O_i^{\text{ex}} = [o_{i-1}^{\text{delim}}, O_i]$ be the $i$-th observation segment and the previous delimiter together for $i > 1$ and define $O_1^{\text{ex}} = O_1$. Expanding the numerator of the ratio in $r_n(\theta)$, we have

$$p(S_n, x_{\text{test}}|\theta) = p(x_{\text{test}}|S_n, \theta)p(S_n|\theta) \tag{59}$$

$$= \sum_{h_{\text{test}}^{\text{start}}} p(x_{\text{test}}|h_{\text{test}}^{\text{start}}, \theta)p(h_{\text{test}}^{\text{start}}|S_n, \theta)p(o_n^{\text{delim}}|O_{1:n}^{\text{ex}}, \theta)\prod_{i=1}^{n} p(O_i^{\text{ex}}|O_{1:i-1}^{\text{ex}}, \theta) \tag{60}$$

$$= \sum_{h_{\text{test}}^{\text{start}}} p(x_{\text{test}}|h_{\text{test}}^{\text{start}}, \theta)p(h_{\text{test}}^{\text{start}}|S_n, \theta) \tag{61}$$

$$\sum_{h_n^{\text{delim}} \in \mathcal{D}} p(o_n^{\text{delim}}|h_n^{\text{delim}})p(h_n^{\text{delim}}|O_{1:n}^{\text{ex}}, \theta)\prod_{i=1}^{n} \sum_{h_{i-1}^{\text{delim}} \in \mathcal{D}} p(O_i|h_{i-1}^{\text{delim}}, \theta)p(h_{i-1}^{\text{delim}}|O_{1:i-1}^{\text{ex}}, \theta)$$

$$\tag{62}$$

$$= \sum_{h_{\text{test}}^{\text{start}}} p(x_{\text{test}}|h_{\text{test}}^{\text{start}}, \theta)p(h_{\text{test}}^{\text{start}}|S_n, \theta) \tag{63}$$

$$\sum_{h_n^{\text{delim}} \in \mathcal{D}} p(h_n^{\text{delim}}|O_{1:n}^{\text{ex}}, \theta)\prod_{i=1}^{n} \sum_{h_{i-1}^{\text{delim}} \in \mathcal{D}} p(O_i|h_{i-1}^{\text{delim}}, \theta)p(h_{i-1}^{\text{delim}}|O_{1:i-1}^{\text{ex}}, \theta) \tag{64}$$

$$= \sum_{h_{\text{test}}^{\text{start}}} p(x_{\text{test}}|h_{\text{test}}^{\text{start}}, \theta)p(h_{\text{test}}^{\text{start}}|S_n, \theta)\prod_{i=1}^{n} \sum_{h_{i-1}^{\text{delim}} \in \mathcal{D}} p(O_i|h_{i-1}^{\text{delim}}, \theta)p(h_{i-1}^{\text{delim}}|O_{1:i-1}^{\text{ex}}, \theta) \tag{65}$$

Note that in the last line, the inner sum is over the set of delimiter states $\mathcal{D}$ by using the assumption that observing a delimiter $o^{\text{delim}}$ implies that the corresponding hidden state $h^{\text{delim}}$ must be in $\mathcal{D}$. We also see that $\sum_{h_n^{\text{delim}}} p(h_n^{\text{delim}}|O_{1:n}^{\text{ex}}, \theta) = 1$.

We restrict our attention to $\theta$ where $p(S_n, x_{\text{test}}|\theta) > 0$, since otherwise $\theta$ does not affect the prediction. Expanding $r_n(\theta)$, we have the following upper bound:

$$r_n(\theta) = \frac{1}{n}\left(\log\frac{p(S_n, x_{\text{test}}|\theta)}{p(S_n, x_{\text{test}}|\theta^*)}\right) \tag{66}$$

$$= \frac{1}{n}\left(\log\frac{\sum_{h_{\text{test}}^{\text{start}}} p(x_{\text{test}}|h_{\text{test}}^{\text{start}}, \theta)p(h_{\text{test}}^{\text{start}}|S_n, \theta)}{\sum_{h_{\text{test}}^{\text{start}}} p(x_{\text{test}}|h_{\text{test}}^{\text{start}}, \theta^*)p(h_{\text{test}}^{\text{start}}|S_n, \theta^*)} + \sum_{i=1}^{n}\log\frac{\sum_{h_{i-1}^{\text{delim}} \in \mathcal{D}} p(O_i|h_{i-1}^{\text{delim}}, \theta)p(h_{i-1}^{\text{delim}}|O_{1:i-1}^{\text{ex}}, \theta)}{\sum_{h_{i-1}^{\text{delim}} \in \mathcal{D}} p(O_i|h_{i-1}^{\text{delim}}, \theta^*)p(h_{i-1}^{\text{delim}}|O_{1:i-1}^{\text{ex}}, \theta^*)}\right) \tag{67}$$

$$\leq \frac{1}{n}\left(\log\frac{\sum_{h_{\text{test}}^{\text{start}}} 1 \cdot p(h_{\text{test}}^{\text{start}}|S_n, \theta)}{\sum_{h_{\text{test}}^{\text{start}}} c_7 \cdot p(h_{\text{test}}^{\text{start}}|S_n, \theta^*)} + n(\log(c_2) - \log(c_1)) + \sum_{i=1}^{n}\log\frac{\sum_{h_{i-1}^{\text{delim}} \in \mathcal{D}} p(O_i|h_{i-1}^{\text{delim}}, \theta)}{\sum_{h_{i-1}^{\text{delim}} \in \mathcal{D}} p(O_i|h_{i-1}^{\text{delim}}, \theta^*)}\right) \tag{68}$$

$$= \frac{1}{n}\left(-\log(c_7) + n(\log(c_2) - \log(c_1)) + \sum_{i=1}^{n}\log\frac{\sum_{h_{i-1}^{\text{delim}} \in \mathcal{D}} p(O_i|h_{i-1}^{\text{delim}}, \theta)}{\sum_{h_{i-1}^{\text{delim}} \in \mathcal{D}} p(O_i|h_{i-1}^{\text{delim}}, \theta^*)}\right) \tag{69}$$

In the above steps, we used both Propositions 1 and 2 in the terms involving $c_2, c_1$ (bounding the probability of $h^{\text{delim}}$ hidden states) and $c_7$ (bounding the probability of $x_{\text{test}}$). Note that in the second line, the sum can must be over the set of delimiter states $\mathcal{D}$ by using the assumption that observing a delimiter $o^{\text{delim}}$ implies that the corresponding hidden state $h^{\text{delim}}$ must be in $\mathcal{D}$.

Focusing on the numerator of the ratio term and summing over the start hidden state for the $i$-th example,

$$\sum_{h_{i-1}^{\text{delim}}\in\mathcal{D}}p(O_i|h_{i-1}^{\text{delim}},\theta)=\sum_{h_{i-1}^{\text{delim}}\in\mathcal{D}}\sum_{h_i^{\text{start}}}p(O_i|h_i^{\text{start}},\theta)p(h_i^{\text{start}}|h_{i-1}^{\text{delim}},\theta)) \tag{70}$$

$$=\sum_{h_i^{\text{start}}}p(O_i|h_i^{\text{start}},\theta)p(h_i^{\text{start}}|\theta)\sum_{h_{i-1}^{\text{delim}}\in\mathcal{D}}\frac{p(h_i^{\text{start}}|h_{i-1}^{\text{delim}},\theta)}{p(h_i^{\text{start}}|\theta)} \tag{71}$$

$$=\sum_{h_i^{\text{start}}}p(O_i|h_i^{\text{start}},\theta)p(h_i^{\text{start}}|\theta)\sum_{h_{i-1}^{\text{delim}}\in\mathcal{D}}\frac{p(h_{i-1}^{\text{delim}}|h_i^{\text{start}},\theta)}{p(h_{i-1}^{\text{delim}}|\theta)} \tag{72}$$

where the last step applies Bayes' rule. We can lower and upper bound the following quantity for any $\theta$ using Assumption 2:

$$\frac{p(h_{i-1}^{\text{delim}}|h_i^{\text{start}},\theta)}{p(h_{i-1}^{\text{delim}}|\theta)}\leq\frac{p(h_{i-1}^{\text{delim}}|h_i^{\text{start}},\theta)}{c_3} \tag{73}$$

$$\frac{p(h_{i-1}^{\text{delim}}|h_i^{\text{start}},\theta)}{p(h_{i-1}^{\text{delim}}|\theta)}\geq\frac{p(h_{i-1}^{\text{delim}}|h_i^{\text{start}},\theta)}{c_4}. \tag{74}$$

This implies that

$$\sum_{h_{i-1}^{\text{delim}}\in\mathcal{D}}\frac{p(h_{i-1}^{\text{delim}}|h_i^{\text{start}},\theta)}{p(h_{i-1}^{\text{delim}}|\theta)}\leq\frac{1}{c_3} \tag{75}$$

$$\sum_{h_{i-1}^{\text{delim}}\in\mathcal{D}}\frac{p(h_{i-1}^{\text{delim}}|h_i^{\text{start}},\theta)}{p(h_{i-1}^{\text{delim}}|\theta)}\geq\frac{1}{c_4}. \tag{76}$$

Plugging in these bounds, we have

$$r_n(\theta)\leq\frac{1}{n}\left(-\log(c_7)+2n(\log(c_2)-\log(c_1))+n(\log(c_4)-\log(c_3))+\sum_{i=1}^{n}\log\frac{\sum_{h_i^{\text{start}}}p(O_i|h_i^{\text{start}},\theta)p(h_i^{\text{start}}|\theta)}{\sum_{h_i^{\text{start}}}p(O_i|h_i^{\text{start}},\theta)p(h_i^{\text{start}}|\theta^*)}\right) \tag{77}$$

$$=\frac{1}{n}\left(-\log(c_7)+2n(\log(c_2)-\log(c_1))+n(\log(c_4)-\log(c_3))+\sum_{i=1}^{n}\log\frac{p(O_i|\theta)}{p(O_i|\theta^*)}\right) \tag{78}$$

$$\to_{n\to\infty}\mathbb{E}_{O\sim p_{\text{prompt}}}\left[\log\frac{p(O|\theta)}{p(O|\theta^*)}\right]+\epsilon_{\text{delim}}^{\theta} \tag{79}$$

where we set

$$\epsilon_{\text{delim}}^{\theta}=2(\log(c_2)-\log(c_1))+\log(c_4)-\log(c_3). \tag{80}$$

Next, we convert the expectation in the bound into a KL divergence. We have

$$\mathbb{E}_{O\sim p_{\text{prompt}}}\left[\log\frac{p(O|\theta)}{p(O|\theta^*)}\right]=\mathbb{E}_{O\sim p_{\text{prompt}}}\left[\log\frac{p(O|\theta)}{p_{\text{prompt}}(O)}+\log\frac{p_{\text{prompt}}(O)}{p(O|\theta^*)}\right] \tag{81}$$

$$=KL(p_{\text{prompt}}\|p(\cdot|\theta^*))-KL(p_{\text{prompt}}\|p(\cdot|\theta)). \tag{82}$$

We will upper bound the first KL term:

$$KL(p_{\text{prompt}}\|p(\cdot|\theta^*))=\mathbb{E}_{O\sim p_{\text{prompt}}}\left[\log\frac{p_{\text{prompt}}(O)}{p(O|\theta^*)}\right]. \tag{83}$$

Expanding the numerator and denominator of the ratio inside, we have

$$p_{\text{prompt}}(O)=\sum_{H}p_{\text{prompt}}(H[1])p(O[1]|H[1],\theta^*)\prod_{j=2}^{k}p(O[j]|H[j],\theta^*)p(H[j]|H[j-1],\theta^*) \tag{84}$$

$$p(O|\theta^*)=\sum_{H}p(H[1]|\theta^*)p(O[1]|H[1],\theta^*)\prod_{j=2}^{k}p(O[j]|H[j],\theta^*)p(H[j]|H[j-1],\theta^*) \tag{85}$$

which differ in only the hidden start distribution. Using Assumption 5, we have that $p(h|\theta^*) \geq c_8$ for any $h \in \mathcal{H}$, which implies that

$$\frac{p_{\text{prompt}}(h)}{p(h|\theta^*)} \leq \frac{1}{c_8} \tag{86}$$

$$\implies p_{\text{prompt}}(O) \leq \frac{1}{c_8} p(O|\theta^*). \tag{87}$$

Finally, this implies that the KL term is bounded as

$$KL(p_{\text{prompt}}\|p(\cdot|\theta^*)) \leq -\log(c_8). \tag{88}$$

This term is non-negative since $c_8 \leq 1$.

Aiming to decompose the second KL term into a sum over the $k$ tokens, we write $p_\theta^j(o) = p(O[j] = o|O[1:j-1],\theta)$ and $p_{\text{prompt}}^j(o) = p_{\text{prompt}}(O[j] = o|O[1:j-1])$. We have

$$-KL(p_{\text{prompt}}\|p(\cdot|\theta)) = -\sum_O p_{\text{prompt}}(O)\log\frac{p_{\text{prompt}}(O)}{p(O|\theta)} \tag{89}$$

$$= -\sum_O p_{\text{prompt}}(O)\sum_{j=1}^k \log\frac{p_{\text{prompt}}(O[j]|O[1:j-1]))}{p(O[j]|O[1:j-1],\theta)} \tag{90}$$

$$= -\sum_{j=1}^k \sum_O p_{\text{prompt}}(O)\log\frac{p_{\text{prompt}}(O[j]|O[1:j-1]))}{p(O[j]|O[1:j-1],\theta)} \tag{91}$$

$$= -\sum_{j=1}^k \mathbb{E}_{O[1:j-1]\sim p_{\text{prompt}}}\left[KL(p_{\text{prompt}}^j\|p_\theta^j)\right] \tag{92}$$

Then we have that

$$\lim_{n\to\infty} r_n(\theta) < -\sum_{j=1}^k \mathbb{E}_{O[1:j-1]\sim p_{\text{prompt}}}[KL(p_{\text{prompt}}^j\|p_\theta^j)] + \epsilon_{\text{start}}^\theta + \epsilon_{\text{delim}}^\theta \tag{93}$$

The second term (set $\epsilon_{\text{start}}^\theta = \log(\frac{1}{c_8})$) is an error term that depends on how different the starting prompt distribution $p_{\text{prompt}}$ (which is part of $p_{\text{prompt}}$) is to the pretraining distribution. The third term is an error term that comes from the delimiter transitions. The bound is negative when the sum of KL terms is larger in magnitude than the error terms. Note that as $k$ becomes larger, the number of observations of $\theta^*$ "overpowers" the distracting transitions in the prompt distribution. This condition is equivalent to the disinguishability condition (Condition 1).

By assumption, for $\theta \neq \theta^*$ the Condition 1 holds, and thus

$$\lim_{n\to\infty} \frac{p(S_n, x_{\text{test}}|\theta)}{p(S_n, x_{\text{test}}|\theta^*)} = \lim_{n\to\infty} \exp(n\cdot r_n(\theta)) = 0 \tag{94}$$

since $r_n(\theta)$ has a negative, constant limit. Note that $\exp(n\cdot r_n(\theta^*)) = 1$ for $\theta^*$.

$\square$

# E   NON-DISTINGUISHABLE CASE

When Condition 1 is unsatisfied, Equation 14), gives an upper bound on the sum of KL divergences for the next token distributions given different-length histories. In contrast, the in-context task only measures the accuracy of the last ($k$-th) token. The main challenge is to relate the different-length histories to each other to give a more precise bound for the error on the in-context task (last token).

Before addressing this challenge, we give the following lemma, which leverages the result of Ávila Pires & Szepesvári (2016); Steinwart (2007) to relate a bound on the KL divergence to 0-1 loss.

**Lemma 2.** *Let the set of $\theta$ which does not satisfy Condition 1 to be $\mathcal{B}$. Assume that $KL(p_{prompt}(y_{test}|x_{test})\|p(y_{test}|x_{test},\theta)$ is bounded above for all $\theta$ and that $\theta^*$ minimizes the multiclass logistic risk $L_{CE}(\theta) = -\mathbb{E}_{x_{test}\sim p_{prompt}}[p_{prompt}(y_{test}|x_{test})\log p(y_{test}|x_{test},\theta)]$. If*

$$\mathbb{E}_{x_{test}\sim p_{prompt}}[KL(p_{prompt}(y_{test}|x_{test})\|p(y_{test}|x_{test},\theta))] \le \epsilon_\theta \ \ \text{for all} \ \ \theta \in \mathcal{B}, \tag{95}$$

*then*

$$\lim_{n\to\infty} L_{0\text{-}1}(f_n) \le \inf_f L_{0\text{-}1}(f) + g^{-1}\left(\sup_{\theta\in\mathcal{B}}\epsilon_\theta\right) \tag{96}$$

*where*

$$g(\delta) = \frac{1}{2}((1-\delta)\log(1-\delta) + (1+\delta)\log(1+\delta)) \tag{97}$$

*is a calibration function for the multiclass logistic loss for $\delta \in [0,1]$.*

*Proof.* First, we note that we can study the 0-1 risk of the limiting predictor:

$$\lim_{n\to\infty} L_{0\text{-}1}(f_n) = \lim_{n\to\infty}\mathbb{E}_{x_{\text{test}},y_{\text{test}}\sim p_{\text{prompt}}}[\mathbf{1}[f_n(x_{\text{test}}) \ne y_{\text{test}}]] \tag{98}$$

$$= \mathbb{E}_{x_{\text{test}},y_{\text{test}}\sim p_{\text{prompt}}}[\lim_{n\to\infty}\mathbf{1}[f_n(x_{\text{test}}) \ne y_{\text{test}}]] \ \ \text{(dominated convergence, boundedness of indicator)} \tag{99}$$

$$= \mathbb{E}_{x_{\text{test}},y_{\text{test}}\sim p_{\text{prompt}}}[\mathbf{1}[\lim_{n\to\infty}f_n(x_{\text{test}}) \ne y_{\text{test}}]] \tag{100}$$

where in the last step we use that since the output space of $f_n$ is discrete and the probabilities that the in-context predictor takes an argmax over converges, then for $N$ large enough, $f_N(x_{\text{test}}) = \lim_{n\to\infty} f_n(x_{\text{test}})$.

Note that for every input $x_{\text{test}}$, the limiting in-context learning predictor outputs the argmax of a predictive distribution which can be a mixture of predictive distributions over $\mathcal{B}$:

$$\lim_{n\to\infty} f_n(x_{\text{test}}) = \underset{y}{\operatorname{argmax}}\,\mathbb{E}_{\theta\sim q}[p(y|x_{\text{test}},\theta)] \tag{101}$$

for some distribution $q$ over $\mathcal{B}$. The KL divergence between this mixture and the prompt concept is bounded by the KL divergence of any one $\theta \in \mathcal{B}$, due to the convexity of KL:

$$\mathbb{E}_{x_{\text{test}}\sim p_{\text{prompt}}}[KL(p_{\text{prompt}}(y|x_{\text{test}})\|\mathbb{E}_{\theta\sim q}[p(y|x_{\text{test}},\theta)]] \tag{102}$$

$$\le \mathbb{E}_{x_{\text{test}}\sim p_{\text{prompt}}}[\mathbb{E}_{\theta\sim q}[KL(p_{\text{prompt}}(y|x_{\text{test}})\|p(y|x_{\text{test}},\theta))]] \tag{103}$$

$$= \mathbb{E}_{\theta\sim q}[\mathbb{E}_{x_{\text{test}}\sim p_{\text{prompt}}}[KL(p_{\text{prompt}}(y|x_{\text{test}})\|p(y|x_{\text{test}},\theta))]] \tag{104}$$

$$\le \sup_{\theta\in\mathcal{B}}\mathbb{E}_{x_{\text{test}}\sim p_{\text{prompt}}}[KL(p_{\text{prompt}}(y|x_{\text{test}})\|p(y|x_{\text{test}},\theta))] \tag{105}$$

where we can exchange the order of expectations since the KL is bounded (dominated convergence).

From the KL bound $KL(p_{\text{prompt}}(y_{\text{test}}|x_{\text{test}})\|p(y_{\text{test}}|x_{\text{test}},\theta)$, we thus have

$$\mathbb{E}_{x_{\text{test}}\sim p_{\text{prompt}}}[KL(p_{\text{prompt}}(y_{\text{test}}|x_{\text{test}})\|p(y_{\text{test}}|x_{\text{test}},\theta))] = L_{\text{CE}}(\theta) - L_{\text{CE}}(\theta^*) \le \sup_{\theta\in\mathcal{B}}\epsilon_\theta \tag{106}$$

where $L_{\text{CE}}(\theta) = -\mathbb{E}_{x_{\text{test}}\sim p_{\text{prompt}}}[p_{\text{prompt}}(y_{\text{test}}|x_{\text{test}})\log p(y_{\text{test}}|x_{\text{test}},\theta)]$ is the multiclass logistic risk, and $L_{\text{CE}}(\theta^*)$ is the optimal risk over $\theta \in \Theta$ by assumption. Applying Theorem 2.2 and 5.11 of Ávila Pires & Szepesvári (2016), $g$ is a calibration function for the multiclass logistic loss, and allows us to convert the surrogate risk bound to a bound on the 0-1 loss, giving the result. Note that we have zero approximation error here, since $\theta^* \in \Theta$. □

Note that $g^{-1}$ is roughly linear in $\epsilon$ for $\epsilon$ smaller than 0.7, where the bound is non-vacuous.

### E.1 PROOF OF THEOREM 2

*Proof.* By the continuity assumption, we have for any $\theta$ in $\mathcal{B}$ that

$$\sum_{j=2}^{k} KL_j(\theta^* \| \theta) \geq \frac{1}{2} \sum_{j=2}^{k} (\theta - \theta^*)^\top I_{j,\theta^*} (\theta - \theta^*) + (k-1)O(\|\theta - \theta^*\|^3) \tag{107}$$

$$\geq \frac{1}{2}(k-1)\lambda_{\min}(I_{j,\theta^*})\|\theta - \theta^*\|^2 \tag{108}$$

$$\implies \|\theta - \theta^*\|^2 \leq \frac{\epsilon_{\text{start}}^\theta + \epsilon_{\text{delim}}^\theta}{\frac{1}{2}(k-1)(\min_j \lambda_{\min}(I_{j,\theta^*}))}. \tag{109}$$

We use this to bound the last KL term by plugging it in below:

$$KL_k(\theta^* \| \theta) = \frac{1}{2}(\theta - \theta^*)^\top I_{j,\theta^*}(\theta - \theta^*) + O(\|\theta - \theta^*\|^3) \tag{110}$$

$$\leq \frac{1}{2}(\max_j \lambda_{\max}(I_{j,\theta^*}))\|\theta - \theta^*\|^2 + O(\|\theta - \theta^*\|^2) \tag{111}$$

$$\leq \frac{(\epsilon_{\text{start}}^\theta + \epsilon_{\text{delim}}^\theta)(\max_j \lambda_{\max}(I_{j,\theta^*}) + O(1))}{(k-1)\min_j \lambda_{\min}(I_{j,\theta^*})}. \tag{112}$$

Rearranging and noting that $KL_k(\theta^* \| \theta) = \mathbb{E}_{x_{\text{test}} \sim p_{\text{prompt}}}[KL(p_{\text{prompt}}(y_{\text{test}}|x_{\text{test}}) \| p(y_{\text{test}}|x_{\text{test}}, \theta))]$, we have

$$\mathbb{E}_{x_{\text{test}} \sim p_{\text{prompt}}}[KL(p_{\text{prompt}}(y_{\text{test}}|x_{\text{test}}) \| p(y_{\text{test}}|x_{\text{test}},\theta))] \leq \frac{(\epsilon_{\text{start}}^\theta + \epsilon_{\text{delim}}^\theta)(\max_j \lambda_{\max}(I_{j,\theta^*}) + O(1))}{(k-1)\min_j \lambda_{\min}(I_{j,\theta^*})} \tag{113}$$

Plugging into Lemma 2 gives the result. □

### E.2 PROOF OF THEOREM 3

Note that Condition 1 ensures that the sum of KL divergences between positions within a $k$-length input is bounded. This means that we have a bound over not only the last-position KL divergence, but also for all the intermediate tokens. Intuitively, the random length test example allows the in-context predictor to "take credit" for fitting the intermediate tokens. The proof is immediate given the KL bound and Lemma 2, given that the length of $x_{\text{test}}$ is uniformly random between 2 to $k$.

*Proof.* Let the set of $\theta$ that does not satisfy Condition 1 to be $\mathcal{B}$. We have for any $\theta$ in $\mathcal{B}$ that

$$\mathbb{E}_{x_{\text{test}} \sim p_{\text{prompt}}}[KL(p_{\text{prompt}}(y_{\text{test}}|x_{\text{test}}) \| p(y_{\text{test}}|x_{\text{test}},\theta))] \tag{114}$$

$$\leq \frac{1}{k-1} \sum_{j=2}^{k} \mathbb{E}_{O[1:j-1] \sim p_{\text{prompt}}} KL(p_{\text{prompt}}(O[j]|O[1:j-1]) \| p(O[j]|O[1:j-1],\theta)) \tag{115}$$

$$\leq \frac{\sup_\theta(\epsilon_{\text{start}}^\theta + \epsilon_{\text{delim}}^\theta)}{k-1} \tag{116}$$

by Theorem 1 and Condition 1. Plugging this into Lemma 2 gives the result. □

## F EXPERIMENTAL DETAILS

### F.1 GINC DATASET

**Pretraining distribution.** We consider a pretraining distribution from a mixture of HMMs with an interpretable hidden state structure and emission distribution. The HMM hidden state $h_t = [s_t, v_t]$ at time $t$ is composed of an *entity* $v_t \in \{1,...,|\mathcal{V}|\}$ (e.g., Einstein) and a *property* $s_t \in \{1,...,|\mathcal{S}|\}$ (e.g., nationality, first name, last name, other grammatical tokens). We model the entities and properties as independent Markov chains (i.e., a factorial HMM (Ghahramani & Jordan, 1997)), while the emissions depend on both. In pretraining documents, we expect that the entities (e.g., Einstein) change slowly over time while and the properties of the entity (e.g., their nationality) change quickly with some pattern to generate

**Pretraining document**

```
f / h x ax o a k au ap /
a o u au ae f ao an / ah
u y as a k au j w ax l
aw r ae au g au ap / / u
aj ae d a h x af u aj i
r j w j as y x n i ap
```

...

**In-context Prompt**

```
l aw ac / ax aj ae / ac j
```

Figure 8: Example pretraining document snippet (**Left**) and example prompt with 3 training examples, 1 test example, and example length 3 (**Right**). The delimiter token is the backslash.

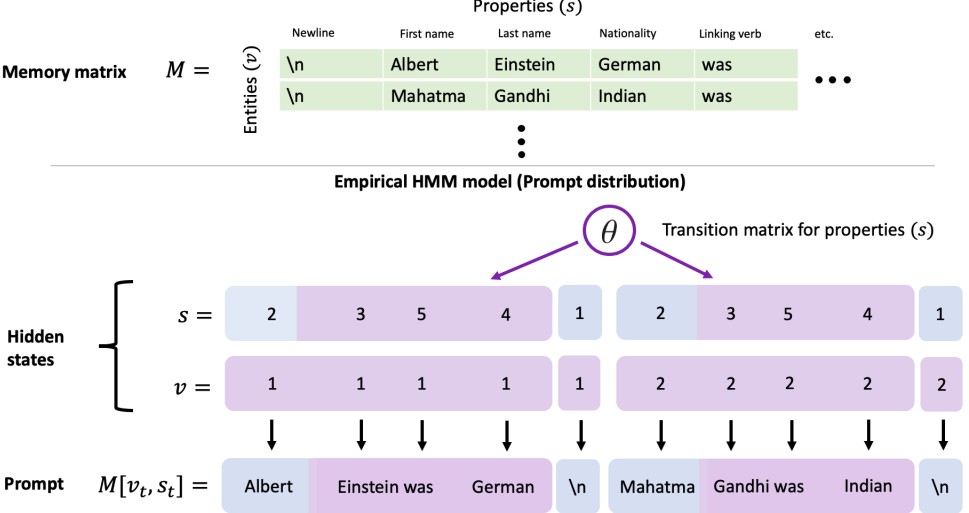

Figure 9: The GINC dataset generates sequences from a mixture of HMMs. The HMM hidden states consist of entities ($v$) and properties ($s$), which index into a memory matrix to produce the observed token. The entity and property sequences are sampled from independent Markov chains. The concept parameter $\theta$ is the transition matrix for properties, which defines relations between properties. In this example, the sequence of properties [2,3,5,4] relates names to nationalities, defining the in-context task. The blue color represents hidden states/observations sampled from the prompt distribution, and the purple color represents hidden states/observations sampled from the pretraining distribution.

natural sentences. We implement this by ensuring that the probability of transitioning to the same entity index in the next step is at least 0.9. The emission distribution depends on a memory matrix $M$ with $|\mathcal{V}|$ rows and $|\mathcal{S}|$ columns (Figure 9). At step $t$, we use the entity $v_t$ and property $s_t$ to index into the memory matrix. In particular, the observed tokens are deterministic with $p(o_t|h_t) = 1$ if $o_t = M[v_t, s_t]$. This construction satisfies the structure on delimiter states (Assumption 1). We ensure that all the transitions have nonzero probability and use a uniform prior over concepts, satisfying Assumptions 2 and 5.

**Concept parameter.** The concept parameter is the property transition matrix, while the entity transition matrix is fixed for all concepts. The prompt start distribution and the concept together determine the in-context task. We define a uniform mixture of HMMs over a family $\Theta$ of 5 concepts to generate 1000 documents with $\sim$10 million tokens total.

**Vocabulary.** The GINC dataset is generated from a mixture of HMMs. These HMMs output tokens from a vocabulary of size in $\{50, 100, 150\}$. The vocabulary contains a special delimiter token (backslash

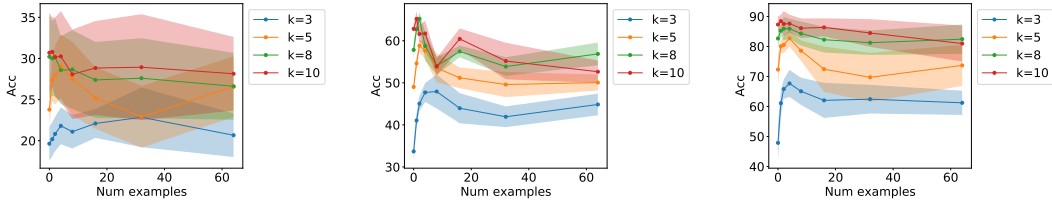

Figure 10: In-context accuracy curve of the 4 layer Transformer on the GINC dataset when the entity transition matrix does not have an additional identity component, for vocabulary sizes 50 (left), 100 (middle), and 150 (right). In-context learning is still generally successful.

– see Figure 8, designated to be index 1. The vocabulary is generated as combinations of letters starting from a to z, then aa to az, and so on. All sequences are tokenized by splitting on whitespaces.

**Memory matrix.** The shared memory matrix has 10 entities and 10 properties, totaling 100 entries (corresponding to 100 hidden states). The first column of the memory matrix is fixed to be the delimiter token, while each remaining entry of the shared memory matrix is populated with a token sampled uniformly from the vocabulary.

**Transition matrix for properties.** We generate 5 property transition matrices, one for each component of the HMM mixture. We generate each transition matrix via a convex combination of 100 random permutation matrices. The weights of the convex combination are randomly generated as

$$\text{softmax}((u-0.5)/t) \tag{117}$$

where $u \in \mathbb{R}^{100}$ has uniform random entries in $[0,1]$ and $t$ is a temperature parameter, set to $0.1$.

**Transition matrix for entities.** The entity transition matrix is shared between all the HMMs that consistute the mixture. The entity transition matrix is generated in the same way as the property transition matrices, except with one additional step. Letting $T$ be a transition matrix sampled in the same way as a property transition matrix,

In pretraining documents, we expect that the entities (e.g., Einstein) change slowly over time while and the properties of the entity (e.g., their occupation) change quickly with some pattern to generate natural sentences. We implement this by ensuring that the probability of transitioning to the same entity index in the next step is at least $0.9$. The final entity transition matrix is then $0.1T + 0.9I$ where $I$ is the identity matrix. Although we add the diagonal component for added realism, we also consider not adding this component. Figure 10 shows in-context learning curves for a small (4 layer) Transformer trained on data that does not add the diagonal component (we check this for vocabulary sizes 50, 100, and 150). In-context learning still works in this case, although not as well for the 50 vocab size case.

**Start distribution.** The starting distribution for the hidden states in all HMMs in the mixture are close to uniform. We generate the start distribution as $\text{softmax}((u-0.5)/t)$ for random vector $u$ with entries uniformly from $[0,1]$ and temperature $t = 10$. In the pretraining documents, we only sample from the start distribution in the beginning of the document.

**Prompt distribution.** To generate the prompts, we first sample a concept $\theta$ uniformly at random from $\Theta$ (well-specification, Assumption 4), then use it to generate all the prompt examples. The prompt start distribution is chosen to be uniform over entities but with a fixed starting property that is chosen randomly for each prompt, for consistency in the task. This may not satisfy Assumption 3, but we found this to still work empirically and is simpler. Given the starting property, we sample $k$ tokens from the HMM defined by the concept $\theta$. Finally, we append the delimiter token for the example. We repeat this process for each example in the prompt, concatenating all examples. The label is generated as

$$\underset{y}{\text{argmax}} \; p_{\text{prompt}}(y|x_{\text{test}}) \tag{118}$$

under the prompt concept $\theta^*$. This differs from the theory, which samples $y_{\text{test}}$ instead of taking it to be the most likely token. However, there can be a large amount of intrinsic error that sampling introduces. We define the label this way in the simulations to remove the intrinsic error from sampling.

**Example of prompt generation.** In the example in Figure 8 (right), the starting property is fixed to be 5 (for example). The first token (l) is generated by sampling a random entity index (3), and indexing into the memory matrix returns l. Running the hidden state chain of the HMM forward gives the next pair of property and entity. Since the entity Markov chain changes slowly, the entity is still 3 in the next step – however, the property has changed to 4, and indexing into the memory matrix outputs the next token (aw). Following this same process to generate the third token (the output for the first example), we finish generating one example. To end the example, we append a delimiter (backslash). We repeat this example generation process for all the examples, except for the test example at the end, where we do not generate the last token. We condition the HMM on the generated prompt to compute the posterior distribution over the next token $p_{\text{prompt}}(y|x_{\text{test}})$. We take the argmax of this distribution to be the ground truth label.

**Dataset details.** The dataset contains 1000 training documents and 100 validation documents, where training documents have 10240 tokens and validation documents have 1024 tokens. Each document is generated by first selecting one of the HMMs from the mixture uniformly at random, then generating 10240 tokens from the HMM.

We also generate 2500 in-context prompts for each (example length,number of examples) pair, for example lengths $k = [3,5,8,10]$ and number of examples $n = [0,1,2,4,8,16,32,64]$. Each prompt is generated using a random HMM in the mixture.

### F.2 TRANSFORMER DETAILS

Our Transformer models are based on the GPT-2 architectures with 4, 12, and 16 layers respectively, with 12 attention heads, 768 dimensional embeddings, residual/embedding/attention dropout set to 0.1, and a context window of 1024. Other than the number of layers, the other parameters are the default settings from the HuggingFace library (Wolf et al., 2019). We train for 5 epochs using the AdamW optimizer (Loshchilov & Hutter, 2019; Kingma & Ba, 2015) with a batch size of 8 and a linear learning rate schedule (with 1000 step warmup) up to a learning rate of 8e-4 for the 4 layer and 12 layer model, while for the 16 layer model we start with a constant learning rate of 8e-4 and reduce by a factor of 0.25 whenever the best validation loss does not improve. We tried both learning rate strategies for all models and take the most consistent. We tuned these models so that the training loss curves between seeds have smaller variability between the runs in terms of the curve shape and when the loss decreases – we found that this is an important indication of stable results. The models took 50 minutes, 2 hours, 3 hours to train respectively. The hardware was mainly Titan Xp GPUs, trained and evaluated using 16-bit precision. All the results are reported with 5 pretraining runs (5 different seeds).

### F.3 LSTM DETAILS

We train an LSTM language model with embedding size 768, hidden layer size 768, and 6 layers. We use dropout 0.2 and weight decay 1e-5. The optimizer is AdamW starting with a learning rate of 1e-3, then reducing by a factor of 0.25 whenever the best validation loss does not go down. We train for a total of 10 epochs, with gradient clipping at norm 1.0. We use a batch size of 8 and backpropagate through time for 1024 steps (each pretraining data segment is also 1024 tokens). Each model takes roughly 2 hours to train on Titan Xp GPUs.

### F.4 VARYING THE VOCABULARY SIZE

To do well on the in-context learning task, the model must both infer the prompt concept and the last HMM hidden state. In general, increasing the number of observable symbols makes the in-context task easier by making the inference of the HMM hidden state easier. With more symbols, each hidden state is more likely to output a different symbol, making the inference problem easier. This improvement comes despite the number of output classes in the problem (same as the vocabulary size) increasing. Figures 11, 12, 13, 14 show in-context learning curves for vocabulary sizes 50, 100, and 150, keeping other hyperparmeters of the dataset the same.

### F.5 EXPERIMENT ON GPT-3

We conduct an additional experiment which shows that longer examples improve in-context learning in GPT-3 on the LAMBADA (Paperno et al., 2016) completion task.

**Data.** In this experiment, we define a short version of the LAMBADA test dataset (LAMBADA test-short) which contains only test examples with up to 200–300 characters in length. We also define two "training" datasets from which to sample examples for the in-context prompts from. The short training dataset (LAMBADA train-short) contains examples from the training set that are

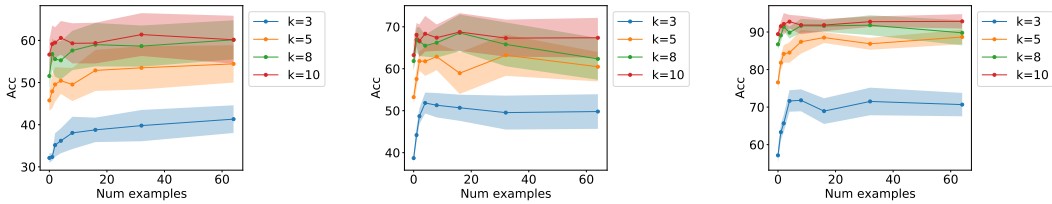

Figure 11: In-context accuracy of the 4 layer Transformer on the GINC dataset for vocabulary sizes 50 (left), 100 (middle) and 150 (right). Accuracies generally improve as the vocabulary size increases.

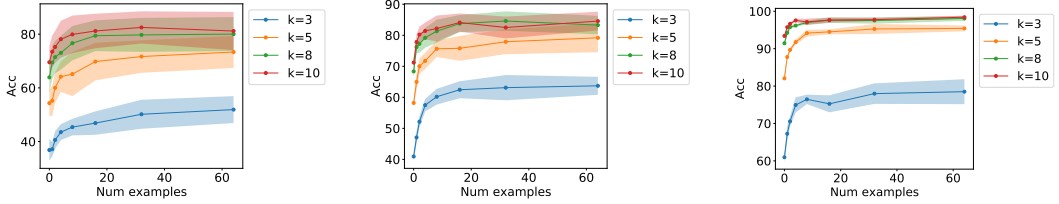

Figure 12: In-context accuracy of the 12 layer Transformer on the GINC dataset for vocabulary sizes 50 (left), 100 (middle) and 150 (right). Accuracies generally improve as the vocabulary size increases.

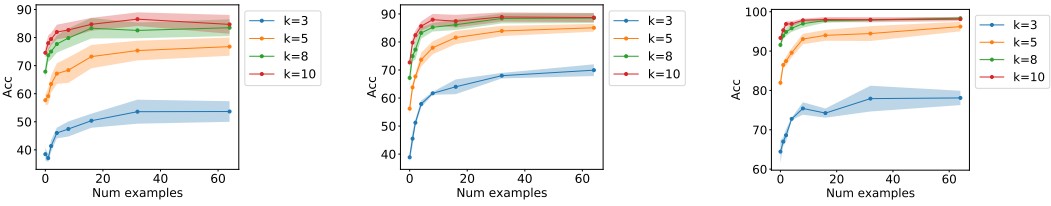

Figure 13: In-context accuracy of the 16 layer Transformer on the GINC dataset for vocabulary sizes 50 (left), 100 (middle) and 150 (right). Accuracies generally improve as the vocabulary size increases.

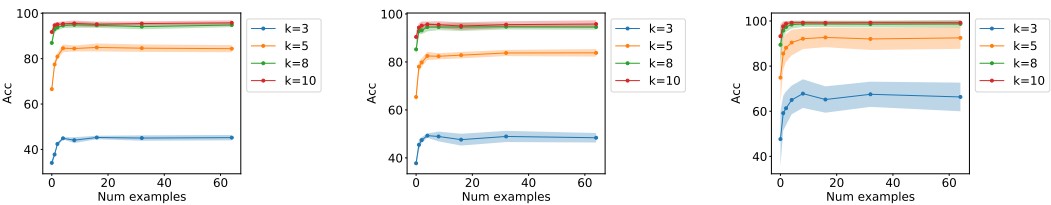

Figure 14: In-context accuracy of the LSTM on the GINC dataset for vocabulary sizes 50 (left), 100 (middle) and 150 (right). Accuracies generally improve as the vocabulary size increases.

200–300 characters in length, which matches the distribution of test-short. The long training dataset (LAMBADA train-long) contains training examples that are 500–600 characters long. We cut the number of examples in the larger of the two training datasets so that the two training datasets are equally sized (47 examples). For each test example, we sample 5 random training examples (5-shot learning).

We also consider equalizing the total length of the prompts in two ways. First, we consider duplicating the 5 short examples (if the examples are [1,2,3,4,5], duplicating refers to [1,2,3,4,5,1,2,3,4,5]). This allows for equalizing the total length without increasing the number of examples. As a skyline comparison, we also consider sampling 10 independent short examples, which contains more input-output pairs for the task.

**Result.** Table 1 shows that when evaluating only on LAMBADA test-short, 5-shot in-context learning using LAMBADA train-long improves the test accuracy by almost 1% compared to LAMBADA train-short, despite the long/short distribution mismatch between train and test. This supports intuitions from our theory.

| Prompt example length | Test Acc (200–300 chars) |
|---|---|
| 5 examples | |
|    Short (200–300 chars) | 69.8 |
|    Long (500–600 chars) | 70.7 |
| 10 examples | |
|    Short, duplicated examples | 69.6 |
|    Short, independent examples | 71.4 |

Table 1: Accuracies for 5-shot in-context learning of GPT-3 on a filtered LAMBADA test set with short examples (200–300 characters). Even though there is distribution mismatch with the test set, having longer examples improves the accuracy, supporting theoretical intuitions. The first two rows use 5 training examples in the prompt, while the last two rows use 10 training examples to equalize the total length.

In comparison, simply increasing the total prompt length by duplicating the short examples does not improve the accuracy. Intuitively, the longer examples have additional information that is not directly related to mapping between the input and output, but can be leveraged to improve in-context learning by helping the model infer the latent concept. Using 5 long examples (as opposed to 5 short examples) closes about 56% of the gap between using 5 short examples and 10 independent short examples despite not adding additional examples or task-related information.

