# OpenReview forum: "An Explanation of In-context Learning as Implicit Bayesian Inference"
_ICLR.cc/2022/Conference — ICLR 2022 Poster_

### Official Review · Reviewer_BHn6 · 2021-10-31

**Correctness:** 4
**Technical Novelty And Significance:** 3
**Empirical Novelty And Significance:** 2
**Recommendation:** 6
**Confidence:** 4

**Main Review:**

__Strengths__: The main strength of the paper is in the connection (and differences) it raises between the ability of ICL to exploit prompts consisting of a sequence of examples of the same type and the Bernstein-Von Mises theorem in Bayesian theory that characterizes the convergence of the posterior to the true underlying parameter, given enough examples.

__Weaknesses__: While the paper goes into mathematical technicalities about certain formal details, it does not explain clearly certain high-level, core, aspects of the approach, which make it difficult for the reader to really understand what is actually claimed. My first question below raises this point in more detail.

__Questions__:

* It is crucial for the reader to clearly understand how the "pretraining distribution" is actually constructed. Although I could not find it detailed in the paper (and actually found some statements that seemed to contradict each other, see below), my understanding is that, after a concept $\theta$ is drawn, a __sequence__ of examples of the concept is produced, separated by a special symbol. The total number of symbols produced is $T$ (is this correct?), a parameter which is not detailed. If this interpretation is correct, am I right to assume that it means that the pretraining distribution, for a given $\theta$, generates essentially the same sequences as the "prompt distribution" for that $\theta$ ?

	* Note: Another interpretation, slightly different from the one I just wrote, would also seem to be supported by the paper, namely that while the pretrained distribution is not _exactly_ the same as the prompt distribution, it does ensure that sequences of examples are generated in almost the same way, but I think the nuance is subtle.

    Assuming my interpretation is correct, the formal part of the paper essentially says that, (under some technical conditions), if we let the pretraining distribution generate a long sequence of examples (but without disclosing which concept $\theta^*$ was employed), and then rerun the (full) pretraining distribution conditionally from that prefix, then the next example generated will tend to be an example "associated" with $\theta^*$.

    If this is the main observation that the paper makes, it should be stated more openly, even if there is a danger that some readers may dismiss it as a bit "obvious" (which it is not really at the formal level).

    * Concerning statements that look to go in different directions relative to this point: the caption in Fig. 1 says "... and are thus 'out of distribution' ", but the example in Fig. 2 seems to concatenate several examples. The paragraph after assumption 3 on p. 4 says: "the prompt is still out of distribution", while the next "regularity" assumption seems to be there to avoid such a possibility...
	* Related to these points, a clearer description of the way the GINC dataset is produced, in terms of whether and how it produces sequences of examples, would be very welcome. Also, the definition of the GINC should be related to the formal assumptions in section 2.1, which it is not.

* Other questions:

    * Please explain why you need a special treatment for the first symbol of each example in the description of the prompt distribution in the first half of p. 3. Is this related with the different color used for "Albert" and "Einstein" in Figure 1 ?
	* In your simulations, the fact that neural LMs such as LSTMs or Transformers are able to recover the completion of examples should be sufficiently explained by their ability to well approximate the pretraining distribution underlying the GINC dataset, without having to invoke additional ICL abilities. However, your experiments indicate that their ability to do completions goes beyond their ability to approximate the underlying distribution. Do you think your theory could be extended to explain that phenomenon, or is it orthogonal to it?

__Comment on the title of the paper__: as a last comment, I feel that the title, claiming to provide an _Explanation_ of iCL in terms of Bayesian learning, exaggerates the current contribution. A true explanation would require to go into how _the hidden states of a pretrained Transformer model_ focus more and more onto a characterization of the task being illustrated by the sequence of examples in the prompt (and typically do so based on only a few examples) and are often able to solve that task even if these examples appear to be remotely related to the training distribution. This is not what the paper does, so I think the title could be reformulated in a more cautious way.

**Summary Of The Paper:**

The paper attempts to bring clarification to In-Context Learning (ICL) by relating it to Bayesian Learning over mixtures of HMMs. Each component of the mixture generates examples of facts of the same conceptual type, such as "Albert Einstein was German", "Marie Curie was Polish", etc (here the conceptual type is "nationality"). The first part of the paper addresses a theoretical question, namely whether the mixture model, on being given a prompt, that is, a prefix such as "Albert Einstein was German. Marie Curie was Polish. Isaac Newton was ", is able to implicitly identify the relevant component of the mixture (i.e. "nationality"), and consequently, whether it is able to correctly complete the prompt, here with "British".
The second part of the paper consists in simulations, where the mixture model is used to produce data on which a neural language model (LSTM or Transformer) is trained, and on evaluations of how well the neural LM is able to perform the completions, under different configurations and ablations.

**Summary Of The Review:**

The paper raises interesting connections between Bayesian learning and certain aspects of in-context learning, based on a simplified formal model exploiting mixtures of HMMs. While the formal developments are extensive, the take-home message is somewhat obscured in the description, making it difficult to assess the true impact of the paper.


*After reading authors' responses*: Thank you for your detailed answers and for the clarifications in the paper. I am not fully convinced by the large significance you put on the OOD nature of the prompts relative to the pretrained distribution --- technically true, but in a narrow and not very illuminating sense IMO ---, and still think your title is an exaggeration. However, I feel that your paper deserves to be discussed by the community and therefore I am raising my score, leaning towards acceptance.

---

> ### Author Response · Authors · 2021-11-18
> **Prompt distribution is different to pretraining distribution and is the basis for novelty of theory**
>
> We thank the reviewer for valuable feedback. BHn6 appreciates the “interesting connections between Bayesian learning and certain aspects of in-context learning” and that the “formal developments are extensive”.
>
> Before we respond to the concerns, we give a high-level overview.
> - **Setup**: ​​We assume that pretraining data distribution is a mixture of HMMs, and the language model learns the pretraining distribution perfectly. Our main result is that we show that this language model can do in-context learning despite the prompts being out-of-distribution (OOD) with respect to the pretraining distribution.
> - **Framework**: At a high level, we prove that this in-context learning can emerge from modeling long-range coherence in the pretraining data. We assume all sentences in a pretraining document share a latent concept (long-range coherence). To generate coherent next tokens in LM pretraining, the LM must infer the latent concept. Inferring a shared concept from examples in a prompt leads to in-context learning.
> - **BHn6 was unclear about why the prompts are OOD**, which leads to the main concern about the significance/novelty of the theoretical results. At a high level, the prompts are OOD due to generating independent examples and concatenating them together, creating unnatural transitions between examples. We clarify this below.
>
> > BHn6 asks whether it is “right to assume that it means that the pretraining distribution, for a given θ, generates essentially the same sequences as the "prompt distribution" for that θ ?”
>
> We thank the reviewer for the important clarifying question -- **we would like to clarify that this is *not* the right interpretation**, and we will clarify this in the paper. The main technical challenge is that **the pretraining distribution and the prompt distribution are not the same, even for a given θ**. We give the reasons for this below.
> - **Why is the prompt OOD?** We illustrate this with a simplified example. Suppose a pretraining document is “a b c d … x y z \n a b c d … y z a b c d ...” where the latent concept enforces the tokens to be in alphabetical order with high probability. A prompt is generated by first **independently** generating examples (e.g., “a b” , “x y” , “f g”) with random starting tokens drawn from a prompt start distribution. The prompt concatenates these examples, separated by a delimiter (“a b \n x y \n f g \n c” ) with a test input “c” at the end. The **transitions between examples can be unnatural** (e.g., “b” rarely transitions to delimiter, “x” rarely follows so closely after “b”). In general, there may not be a concept that places high probability on transitioning to a delimiter token every $k$ tokens. **This means that prompts are usually very low probability (OOD) sequences under the pretraining distribution**. Even though the examples were independently generated under the *prompt distribution*, we can no longer factorize the examples under the *pretraining distribution*, which does not have the factorized structure over examples.
>
>
> > BHn6 asks “The total number of symbols produced (in a prompt) is T (is this correct)”?
>
> This is not correct - the parameter T is the length of a pretraining document, which is not related to the length of the prompts for in-context learning. We’ll make this clear in the paper.
>
> > BHn6 asks “The paragraph after assumption 3 on p. 4 says: "the prompt is still out of distribution", while the next "regularity" assumption seems to be there to avoid such a possibility”?
>
> The **regularity assumptions do not make the prompt distribution and pretraining distribution the same**. They only ensure that the prompts have nonzero probability under the pretraining distribution, which is necessary for the Bayesian framework.
>
> > Clarity on interpretation/novelty of the theory
>
> We thank the reviewer for the feedback. The theory is in a novel setting since the prompts are OOD, and existing theory does not handle this. Details:
> - Previous theory on Bayesian asymptotics (Bernstein von Mises-type results) either assume 1) independence of the observations or 2) a sequential observation drawn from the same HMM distribution (we believe 2 is the setting that the reviewer has interpreted, which is not our setting). For 1), this would typically mean that the pretraining distribution also factorizes over the examples, which is not true here. For 2), existing results can handle the case of observing one long sequence from the pretraining distribution (like a pretraining document), but do not extend to OOD sequences. Our theory deals with these challenges.
> - We thank the reviewer for the suggestion to clarify the description of GINC. We added a better description of GINC in Section 4 and a concrete example in the appendix (Fig 7). We will clarify further with respect to the assumptions in section 2.1 - **GINC is set up to satisfy the formal assumptions**.

---

> > ### Author Response · Authors · 2021-11-18
> > **Author response continued**
> >
> > > BHn6 comments “Please explain why you need a special treatment for the first symbol of each example”.
> >
> > We’ll make sure to clarify this. The **first symbol of each example is sampled from the prompt start distribution, which controls the distribution of inputs**. This allows the prompt distribution to focus on a particular distribution of inputs (such as full names of famous people in Fig 1). **This is another way the prompt distribution can differ from the pretraining distribution**. BHn6 is correct that the different color for “Albert” in Fig 1 denotes that the first symbol “Albert” is drawn from a different distribution (prompt start distribution) to the second symbol “Einstein”.
> >
> > > BHn6 notes that on GINC, Transformers and LSTM’ in-context learning accuracy improves when model size gets bigger despite having the same pretraining loss “(goes beyond their ability to approximate the underlying distribution). Do you think your theory could be extended to explain that phenomenon, or is it orthogonal to it?”
> >
> > This is a great question. Since **our theory focuses on the role of the pretraining distribution, the theory does not cover the role of model size or architecture**. We leave this to future work.
> >
> > > BHn6 comments that “a true explanation would require to go into how the hidden states of a pretrained Transformer model” learn/concentrate on the task and how in-context learning often works even when the examples are only “remotely related to the training distribution”.
> >
> > - While we agree that a full understanding of in-context learning would open the black box of the model internals, we aim to provide an explanation that explicitly focuses on the role of the pretraining distribution on the emergence of in-context learning. In our GINC experiments, we find that **in-context learning occurs in both Transformers and LSTM model architectures**, which suggests that the emergence of **in-context learning we see is not due to any specific hidden state dynamics in a Transformer**, but rather the latent variable structure in the pretraining data.
> > - The theory operates in a setting where the prompts are low-probability/OOD sequences, which is a first step towards characterizing in-context learning for prompts that are not related to the training distribution. We agree that GPT3 seems to exhibit some ability to extrapolate to unnatural/unseen tasks, but we believe the Bayesian framework can be extended to capture this. For example, consider the empirical mixture of HMMs with properties and entities. If both the property and entity transition matrices are latent variables in the pretraining distribution, it would be possible to infer a novel property and entity matrix pair at test time without seeing all the pairs during pretraining, which amounts to extrapolation. We’ll include a discussion of this extension in the paper.
> >
> > > BHn6 comments that for the theory, the “ the take-home message is somewhat obscured in the description”.
> >
> > We thank the reviewer for the feedback. Beyond the above points about theoretical contributions, we offer some additional takeaways:
> > - Conceptually, the **theoretical understanding helps with reasoning about the issues of prompt structure** (how it makes prompts OOD and its effects on in-context accuracy) as well as previously unexplored aspects such as the length of each example. In Section E.7, we run an experiment on GPT3 which studies the effect of example length on LAMBADA. We find that longer train examples improve in-context performance even when the test examples are short, which supports the theory.
> > - We believe our **small-scale GINC dataset can be a testbed for in-context learning**. Many empirical phenomena we see on our GINC dataset (improved performance with number of examples, bigger models are better) are also seen in large LMs [1]. **During rebuttals, we provide additional evidence (Section E.5, E.6) that GINC captures real-world in-context learning phenomena.** In Section E.5, we show that in-context accuracy in GINC is highly dependent on example ordering, similar to GPT-3 results from [2]. In Section E.6, we show that in some settings of GINC, zero-shot can be better than one/few-shot accuracy but accuracy later improves with more examples, which mirrors the behavior of GPT-3 on some datasets (e.g., LAMBADA, HellaSwag, PhysicalQA, RACE-m) [1]. This is possible since one/few-shot prompts introduce the unnatural/OOD transitions between examples.
> > - Because we generate a **synthetic dataset, we can also control the latent properties of the data (e.g., out of distribution concepts) which can make predictions about LLMs** - we predict that completely unseen concepts cannot be in-context learned (but it's possible that the model can learn to combine some learned concepts together).
> >
> > [1] Tom Brown et al. Language models are few-shot learners. 2020
> >
> > [2] Tony Z. Zhao et al. Calibrate Before Use: Improving Few-Shot Performance of Language Models. ICML, 2021.

---

> > > ### Comment · Reviewer_BHn6 · 2021-11-28
> > > **Thank you for your responses and for clarifications. Despite remaining doubts I think the paper deserves to be discussed by the community.**
> > >
> > > Thank you for your detailed answers and for the clarifications in the paper. I am not fully convinced by the large significance you put on the OOD nature of the prompts relative to the pretrained distribution --- technically true, but in a narrow and not very illuminating sense IMO --- and still think your title is an exaggeration. However, I feel that your paper deserves to be discussed by the community and therefore I am raising my score, leaning towards acceptance.

---

> > > > ### Author Response · Authors · 2021-11-29
> > > > **Thanks for the response!**
> > > >
> > > > Thanks for your response! While our setup doesn't capture all the possible ways that prompts can be different from natural language, we focus on a core aspect of the prompt structure - as a list of independent examples separated by delimiters. We hope that future work will build on this to cover the full variety of prompting formats and methods.
> > > >
> > > > In general, we believe the OOD nature of prompts is important because it connects to the main mystery of in-context learning: why can a language model trained on the pretraining distribution do things that it was not trained to do? After all, prompts seem different from natural language due to the special format. For language modeling, this boils down to finding a relation between the pretraining distribution and prompts despite their differences, which is what our theory provides.
> > > >
> > > > We agree that this is a first step in understanding in-context learning. Do you have suggestions on the title? We'd be happy to consider them. One proposal could be to replace the beginning with "A view" or "A characterization".

---

> > > > > ### Comment · Reviewer_BHn6 · 2021-11-29
> > > > > **Thanks for your comments**
> > > > >
> > > > > Thank you for your comments. I do not think it would be appropriate for me to propose a new title for your paper, this is wholly up to you, in case you feel it would be helpful, but thank you for asking. Still, IMO, "A characterization" is not much different from "An explanation", as opposed to "A view".

---

### Official Review · Reviewer_VX98 · 2021-11-03

**Correctness:** 4
**Technical Novelty And Significance:** 3
**Empirical Novelty And Significance:** 3
**Recommendation:** 6
**Confidence:** 3

**Main Review:**

This paper proposes and analyzes a framework under which "in-context learning" can be studied from a statistical perspective. Despite the potential limitations of the framework and assumptions, this is an original perspective that allows for theoretical investigation of a complex empirical phenomenon. My main comments are focused on three topics:

1. It would be useful for the authors to give a more robust account of why we need a specific account of "in-context learning" in the first place. Put differently, why should we not just consider these examples to be relatively straightforward cases of next-word prediction for a model that has learned an accurate approximation to the distribution over sequences of words? In the first paragraph of $\S$1, the authors write, "Given that the distribution of prompts are quite different from real text..." - but is this really true, particularly given the enormous breadth of modern training corpuses? The prompt that precedes the quote seems reasonably close to natural (if somewhat simple) language in this case, certainly enough for a reader to reasonably ask whether there is anything particularly special here that needs explaining. How do we distinguish between "recognition and continuation of a specific concept / pattern / construction" and just "good next-word prediction conditioned on a relatively plausible sequence?"

2. The overall strategy of the analysis is clear, particularly the high-level walkthrough that precedes the theorem statements and proofs. It is also clear that the analysis depends at several points on specific, technical aspects of the problem formalization, including the mixture-of-HMMs model for the pretraining distribution (Eq. 5) and the specific structure of the HMM transitions around delimiter tokens (Eqs. 9-10). The paper could be improved by some discussion of these assumptions. In particular, what is critical to this framework, and what is required merely for technical purposes in some proofs? Why does the mixture-of-HMMs model seem to be the right choice for this analysis?

3. The experiments are carefully documented and results include estimates of uncertainty where appropriate. The prediction performance as a function of $n$ and $k$ generally supports the theoretical results, and the ablations indicate the importance of the specific mixture-of-HMMs structure of the pretraining distribution for this task. However, some aspects of the experimental approach could use more discussion. One main question is: how does this toy setting help us understand the emergence of "in-context learning" in models trained on vast corpuses whose contents are very unlikely to be well-modeled by a mixture of HMMs corresponding to well-separated "concepts?" We see that this phenomenon disappears when the LSTM and transformer models are trained on "random" data, but the real-world data on which language models are trained clearly lives between these extremes of rigid concept structure and meaningless sequences. Some lower-level questions: Does the theory explain why performance seems to be relatively more sensitive to $k$ than $n$? Why do we observe "in-context learning" in these relatively small LSTM and transformer models, when previously it was observed that even models as large as GPT-2 fail to demonstrate this phenomenon?

Other comments:
- The first sentence of $\S$3, particularly "... conditioning the pretraining distribution on the prompt infers the prompt concept to enable in-context learning..." is unclear.
- What are $c_1$ and $c_2$ in Eq. (13)? What is $k$ in Eq. (14)?
- To avoid ambiguity, I'd suggest using some other symbol for the footnote in Thm. 2.
- Typo in first paragraph of $\S$1: "Intuigingly"

**Summary Of The Paper:**

The authors propose to study the phenomenon of "in-context learning" through the lens of Bayesian prediction. They introduce a framework in which the data generating distribution is given as a mixture of HMMs parameterized by a latent "concept," and they show that under suitable conditions the posterior predictive distribution over a completion given a structured prompt asymptotically selects the "concept" behind this particular structure. They illustrate this framework and reproduce the phenomenon in a simplified language modeling example.

**Summary Of The Review:**

This paper contributes a framework for theoretical exposition of a complex empirical behavior observed in language models. This framework and its results offer useful insights, and these are corroborated by a clear set of experiments. There remain some questions as to the motivation for "in-context learning" as a specific phenomenon of interest, the potential limitations of some assumptions that seem critical to the proofs, and the gap between the toy setting and "in-context learning' as it arises in large language models. In my view the strengths outweigh the weaknesses, and my score could potentially be improved with some further discussion of the points above.

---

> ### Author Response · Authors · 2021-11-18
> **Comparison to next-word prediction and critical assumptions of the framework**
>
> We thank the reviewer for the feedback. VX98 thought that the paper provides an “original perspective that allows for theoretical investigation of a complex empirical phenomenon”, “offers useful insights”, and has a “clear set of experiments”.
>
> We respond to the concerns below:
> > VX98 asks “how do we distinguish between "recognition and continuation of a specific concept / pattern / construction" and just "good next-word prediction conditioned on a relatively plausible sequence?””.
>
> We thank the reviewer for the great question. Good next-word prediction does not suffice for in-context learning. For in-context learning, **the model must “learn” what the task is to make the correct prediction**. For example, a good next-word prediction model may output “Marie Curie was brilliant” as a plausible completion, but the in-context learning phenomenon is that the model tends to learn the same latent concept as the examples preceding the test example to output “Marie Curie was Polish”.
>
> > On the theoretical assumptions, VX98 asks “what is critical to this framework, and what is required merely for technical purposes”, and why does the “mixture-of-HMMs model seem to be the right choice for this analysis”?
>
> We thank the reviewer for the great question, and we will clarify this in the paper. At a high level, the **critical component of the framework is the long-range coherence in the pretraining documents**, which comes from the latent concept that is shared across the sentences in the document. Successful language modeling requires inferring this latent concept as an intermediate step, and applying this inference procedure to infer the shared concept across prompt examples leads to in-context learning. **Thus the latent variable structure is core to the framework**. **The mixture of HMMs structure is mainly for tractability of theoretical analysis**, where we leverage the Markov property in the hidden state space. However, we do not expect the in-context learning phenomenon to be special to pretraining distributions based on HMMs.
> - Assumption 3 (well-specification) is possibly not needed, as there are works on Bernstein-von Mises-style results under misspecification [3], and it is possible that ideas from this work can be extended here. However, [3] doesn’t cover our setting, where the prompt examples are not independent with respect to the pretraining distribution. An ideal result under misspecification would be that the model learns to pick the latent concept in the family that is closest in KL divergence to the prompt concept.
> - Assumptions 1,2 (delimiter structure) and Assumption 4 (regularity) are specific to our proof strategy, which revolves around comparing the likelihood of the prompt under two different latent concepts. To compare the likelihoods, the probability of the prompt under the pretraining distribution conditioned on the prompt concept θ∗ should be nonzero (but usually low) - otherwise, the denominator of the ratio is zero. Transition probability bounds in Assumptions 4 and 2 ensure that the prompt has nonzero probability. The delimiter structure is used to approximately factorize the prompt into examples + error coming from the delimiters, which is another part of our specific proof. Practically, delimiters play an important role in allowing the model to separate different examples. Thus, some properties of the delimiter are likely needed to ensure that the model can distinguish different examples.

---

> > ### Author Response · Authors · 2021-11-18
> > **Connection to real data/models and why small-scale is possible**
> >
> > > VX98 asks why we observe in-context learning “in these relatively small LSTM and transformer models, when previously many larger models did not demonstrate this phenomenon.”
> >
> > This is a great question - **this is because the vocabulary size and overall complexity of the pretraining distribution is smaller, and therefore the data and models are relatively larger** compared to the overall complexity, allowing us to observe these phenomena at a smaller absolute scale. We view this as a strength of generating a synthetic dataset where we have full control. We keep dataset hyperparameters such as vocab size, HMM hidden state size, and number of latent concepts relatively small to keep the complexity low.
> >
> > > VX98 asks “how does this toy setting help us understand the emergence of "in-context learning" in models trained on vast corpuses whose contents are very unlikely to be well-modeled by a mixture of HMMs corresponding to well-separated "concepts?"”
> >
> > First, we believe that **showing that in-context learning can come from properties of the pretraining data is an important step** in understanding in-context learning. While real data is unlikely to come from a mixture of HMMs, we hope that our setting captures some of the critical components that contribute to in-context learning, and that our small-scale GINC dataset can be a testbed for in-context learning. **Many of the empirical phenomena that we see on our GINC dataset** (improved performance with number of examples, bigger models are better) **are also seen in large language models** [1]. **During rebuttals, we provide additional evidence (Section E.5, E.6) that GINC captures real-world in-context learning phenomena.** In Section E.5, we show that in-context accuracy in GINC is highly dependent on example ordering, similar to GPT-3 results from [2]. In Section E.6, we show that in some settings of GINC, zero-shot can be better than one/few-shot accuracy but accuracy later improves with more examples, which mirrors the behavior of GPT-3 on some datasets (e.g., LAMBADA, HellaSwag, PhysicalQA, RACE-m) [1]. This is possible since one/few-shot prompts introduce the unnatural/OOD transitions between examples.
> > - Conceptually, the **theoretical understanding helps with reasoning about the issues of prompt structure** (how it makes prompts OOD and how it affects in-context accuracy) as well as previously unexplored aspects such as the length of each example. In the appendix (Section E.7) we run an experiment on GPT3 which studies the effect of example length on the LAMBADA task. We find that longer examples improve in-context performance even when the test examples are short, which supports the theory.
> > - Because we generate a **synthetic dataset, we can also control the latent properties of the data (e.g., out of distribution concepts) which can make predictions about LLMs** - we predict that completely unseen concepts cannot be in-context learned (but it's possible that the model can learn to combine some learned concepts together).
> >
> > > VX98 asks if the “theory can explain why performance seems to be relatively more sensitive to k (the length of each example) than n (the number of examples)” in the experiments?
> >
> > We thank the reviewer for the interesting point. The current theory cannot explain this. For $k$, Theorems 2 and 3 give settings where the error reduces as $1/k$. We consider the case where the number of examples $n$ goes to infinity, but do not characterize the rate with $n$. If the rate is the standard $1/\sqrt{n}$ (slower than $1/k$), this could explain this observation. This is left to future work.
> >
> > > “What are c1 and c2 in Eq. (13)? What is k in Eq. (14)?”
> >
> > c1 and c2 are constants defined in Assumption 2; they are bounds on transition probabilities of delimiter states. k in Eq 14 is the length of each example defined in Section 2 and mentioned in the paragraph after Eq 14. We’ll make this clearer in the paper.
> >
> > [1] Tom Brown et al. Language models are few-shot learners. 2020
> >
> > [2] Tony Z. Zhao, Eric Wallace, Shi Feng, Dan Klein, Sameer Singh. Calibrate Before Use: Improving Few-Shot Performance of Language Models. ICML, 2021.
> >
> > [3] B.J.K. Kleijn and A.W. van der Vaart. The Bernstein-von mises theorem under misspecification. Electronic Journal of Statistics, 6, 2012.

---

> > > ### Comment · Reviewer_VX98 · 2021-11-29
> > > **Reply**
> > >
> > > Thanks to the authors for their detailed and thoughtful discussion of the points raised by all reviewers. Each of the main points in my review has been addressed in the comments above, and particularly in light of the further discussion I believe that this paper offers a perspective and contribution that is worth sharing with the broader community. I will maintain my score and advocate for the paper's acceptance.
> > >
> > > With respect to the claim that prompts are out-of-distribution, I appreciate the further discussion but remain gently skeptical. To me, the fact remains that the example prompt in Fig. 1 is perfectly valid natural language, and this combined with the reasonable proposition that "thematic repetition is common in natural language" seems enough to explain this example of in-context learning simply as "good conditional prediction by a well-trained language model." I don't think this is very well refuted by the new example in $\S 2$, in which the true distribution is so simple we can easily see that the prompt is unfamiliar. The GINC examples are something of a middle ground here, and the demonstration of in-context learning under controlled settings where prompts are guaranteed to be out-of-sample does register as important evidence. There may also be be further experiments that could disambiguate here - for example, using "concept" patterns that are clearly unnatural, and so very probably not in the training set, yet simple to recognize and extend - but this is not necessarily in scope for the current paper.

---

### Official Review · Reviewer_qavU · 2021-11-03

**Correctness:** 3
**Technical Novelty And Significance:** 4
**Empirical Novelty And Significance:** 2
**Recommendation:** 6
**Confidence:** 3

**Main Review:**

I think the paper does a good job in explaining why concatenating independent examples as a sequence and prompting a LM trained on HMMs gives good results, considering the prompt sequence can be OOD. I have some questions about practicality of the approach and the observation generation process for the prompting.

1- I think there is a significant gap between simulated experiments and real sequential data. Showing that the theory holds for the simulated data as the initial step is useful, but HMMs are far from capturing real data and results with existing LMs would be needed to understand the gap. For example, GPT-3 zero-shot results are actually better than GPT-3 one-shot results on LAMBADA and HelloSwag datasets (Table 3.2) [1] and few-shot results are also lower for smaller networks (Figure 3.2).

- There is an additional result in the appendix related to GPT-3 but it is not clear if using longer examples gives better results or because the overall prompt sequence is longer. Can you also present results where you use more short examples that gives the same prompt length as that of longer examples (such as using 10 short vs 5 long examples)?

- GPT-3 uses an additional task description, which would be similar to the concept in your paper, could you explain how this can be incorporated?

2- The example sequence (O_i) is generated using the same pre-trained language model. In this case, OODness of the prompt sequence only comes from the fact that examples are concatenated. I think having prompt distribution not necessarily sharing the same pre-training data distribution could be important.
- Could you train an additional language model on your corpus and generate prompt observations from that?


3- How sensitive is your results to different orders of the same set of examples?

[1] Language Models are Few-Shot Learners. OpenAI.


**Summary Of The Paper:**

This paper studies the question of why few-shot prompting works for large language models (LM) such as GPT? The paper is mainly focused on understanding why prompting works given that concatenating multiple examples separated by a delimiter introduces out-of-distribution prompts that the LM would most likely not have seen during its pre-training. The paper shows that when the pre-training distribution is a mixture of HMMs, prompting works as a result of Bayesian inference. The authors show the validity of their theoretical results by training transformers and LSTM on synthetic datasets sampled from some HMMs that they devised.

**Summary Of The Review:**

I think the theory behind the paper is really interesting to understand in what conditions few-shot prompting might work. I also think that there needs to be more experiments with real data to understand the gap between their simulated experiments and real language.

---

> ### Author Response · Authors · 2021-11-18
> **Connections to real data and new experiments**
>
> We thank the reviewer for the valuable feedback. Reviewer qaVU notes that the “theory behind the paper is really interesting to understand in what conditions few-shot prompting might work” and that the paper “does a good job in explaining why concatenating independent examples” leads to in-context learning “considering the prompt sequence can be OOD”. We address the concerns below:
>
> > qavU notes that there is a “gap between simulated experiments and real sequential data”. In particular, they mention the finding from the GPT3 paper that “ GPT-3 zero-shot results are actually better than GPT-3 one-shot results on LAMBADA and HelloSwag datasets (Table 3.2) [1] and few-shot results are also lower for smaller networks“
>
> In the revision, we provide additional evidence (Section E.5, E.6) that GINC captures in-context learning phenomena in real systems. **In Section E.6, we show that in some settings of GINC, zero-shot can be better than one/few-shot accuracy but accuracy later improves with more examples**, which mirrors the behavior of GPT-3 on some datasets (e.g., LAMBADA, HellaSwag, PhysicalQA, RACE-m) (Brown et al 2020 [1]). This is possible since one/few-shot prompts introduce the unnatural/OOD transitions between examples. **In Section E.5, we show that in-context accuracy in GINC is highly dependent on example ordering**, similar to GPT-3 results from Zhao et al 2021 [2].
>
> >  In the appendix (Section E.7) we run an **experiment on GPT3 which studies the effect of example length** on the LAMBADA task. We find that longer examples improve in-context performance even when the test examples are short, which supports the theory. qavU asks “Can you also present results where you use more short examples that gives the same prompt length as that of longer examples”?
>
> During the rebuttal period, **we ran the LAMBADA GPT-3 example length experiment on two more settings (Section E.7) that equalize the total prompt length** between the short and long example test sets. First, we find that simply duplicating the 5 short examples to double the overall length does not improve in-context accuracy (69.6% vs 69.8% with 5 short examples). Thus, the total prompt length of the 5 long examples does not contribute to the improvement in performance. Second, we find that using 10 independent short examples increases the in-context accuracy (71.4% vs. 69.8%). Thus, **using 5 long examples (70.6%) closes about 56% of the gap between using 5 short examples and 10 independent short examples despite not adding additional examples or explicit task-related information**. Intuitively, the longer examples have additional information that is not directly related to mapping between the input and output, but can be leveraged to improve in-context learning by helping the model infer the latent concept.
>
> > qavU asks “GPT-3 uses an additional task description, which would be similar to the concept in your paper, could you explain how this can be incorporated?”
>
> We thank the reviewer for the great question. Without the task description the latent concept is more challenging to learn, but we could extend the framework to allow for partial observations of the latent concept to model task descriptions. Intuitively, task descriptions would improve the ability of the model to infer the latent concept.
>
> > qavU asks “Having prompt distribution not necessarily sharing the same pre-training data distribution could be important. Could you train an additional language model on your corpus and generate prompt observations from that?”
>
> We agree that considering prompts from unseen concepts is important for assessing extrapolation, and **we do this experiment in Section 4.2**. Here, we consider generating prompts from “out-of-distribution”/unseen concepts that are not in the pretraining distribution. We find that in-context learning does not extrapolate to these random unseen concepts.
>
> > qavU asks “How sensitive is your results to different orders of the same set of examples?”
>
> We thank the reviewer for the question. During rebuttals, we ran an experiment to test the effect of example ordering. **In Section E.5, we show that in-context accuracy in GINC is highly dependent on example ordering, similar to GPT-3 results from Zhao et al 2021 [2].** Although our current theory does not explain this order dependence phenomenon (partially due to taking the number of examples to infinity), we believe that extending the analysis on finite samples and taking into account the effect of strong local dependencies in the pretraining data could capture the effect of example ordering.
>
> [1] Tom Brown et al. Language models are few-shot learners. 2020
>
> [2] Tony Z. Zhao, Eric Wallace, Shi Feng, Dan Klein, Sameer Singh. Calibrate Before Use: Improving Few-Shot Performance of Language Models. ICML, 2021.

---

> ### Author Response · Authors · 2021-12-02
> **Further discussion**
>
> Dear reviewer, may we ask if you could respond to our comments? In our response below, we provide additional experiments on example ordering and zero-shot better than one-shot on GINC and expanded the baselines for the GPT-3 experiment. We believe these strengthen the empirical connections to real data. Please let us know if you have other questions or concerns. Thank you!

---

### Official Review · Reviewer_j2L6 · 2021-11-09

**Correctness:** 3
**Technical Novelty And Significance:** 4
**Empirical Novelty And Significance:** 4
**Recommendation:** 6
**Confidence:** 4

**Main Review:**

### Main Review

#### Strengths

1. Significance: The paper studies a timely question: Why does in-context learning (in particular in LLMs) take place? The small-scale reproduction and theoretical model of this phenomenon may lead to new understanding.

2. Novel empirical phenomenon: The existence proof of in-context learning in small-scale models is new and surprising, to my knowledge.

#### Weaknesses

1. I am not sure of the novelty of the theoretical results. The theoretical model considered in the paper seems to be a mixture Hidden Markov Model (HMM). Helske & Helske (2019) demonstrate that such a model can be expressed as a standard HMM, for which inference is known to be possible and moreover efficient. Given this, it is not clear to me what Theorem 1 adds, which shows that the "in-context predictor" (ie. Bayes) infers the latent mixture variable. Theorems 2 & 3 seem novel to my knowledge, as they tie this inference to the expected 0-1 loss, but further clarification on the significance of these results would be helpful to understand the contribution.

1. Broadly, I think the paper should more clearly distinguish ideal performance (ie. Bayes) from models picked out by their behavior (eg. "in-context predictor," Transformer, LSTM) from datasets (ie. pretraining distribution, downstream task). As examples:
    - In Theorem 1, 2, & 3, Bayes is shown to recover latent variables and make accurate predictions in the mixture HMM; i.e., these theorems describe ideal performance. However, in some cases, this ideal performer is described an "in-context predictor" ("....we have that the in-context predictor infers prompt concept..."). Elsewhere in the paper, "in-context predictor" is used to describe a behavior (ie. learning a task from a small context). Using "in-context predictor" widely in this way is problematic, because stating that the ideal model is an in-context predictor is circular with respect to the claim that in-context predictors can be modeled as a Bayesian ideal (which is one of the central claims of the paper).
    - In Section 4, the pretraining distribution is modelled as a mixture HMM, and data generated from this model is used as training data for scaled-down versions of LLMs . This is a distinct use of the mixture HMM formalism (ie. as a data-generating process) from its use to derive the ideal performance described in Theorems 1-3. This distinction could be made clearer in the paper.

1. The results in Section 4 are claimed to demonstrate that in-context learning is taking place in this small-scale setup. However, if in-context learning just refers to improvement in the number of test examples (ie. the behavioral definition), then these results are quite distinct from the theoretical results. Can the asymptotic performance also be plotted to demonstrate that empirical performance approaches the ideal performance described in Section 3? This would more closely tie the "in-context" phenomenon to the theoretical model.

1. The paper should more clearly discuss scope & limitations, and, in particular, which assumptions in the theoretical results and data model may not match reality. For example, the topic-model/mixture assumption; the enumerable token set; the assumption that the "language model fits this pretraining distribution exactly with enough data and expressivity."

1. The paper is a little light on take-aways for LLMs. It has reproduced the in-context learning phenomenon in smaller models, but there lacks a strong connection to argue that the same mechanisms are at play for larger-scale models.

### Minor comments

#### Minor, specific sections:

- Intro & Figure 1: I think the example in the introduction is better served in the caption of Figure 1, and the introduction should describe just the abstract problem and reference Figure 1.
- "Thus, in-context learning can be thought of as implicit Bayesian inference." I think this should be more explicitly explained as "we model in-context learning as Bayesian inference with this particular HMM."
- "Ideally, the posterior predictive distribution should concentrate" Is this referencing prior result? Or an assumption that would be beneficial for the framework? Explain further.
- "During in-context learning, the prompt is presented as a contiguous sequence rather than IID examples and are thus “out-of-distribution” to the language model." I'm not sure I follow why this fact makes the test prompt OoD.
- "The canonical asymptotic tool in Bayesian methods..." The reason for introducing this should be much more elaborated. What is the purpose of looking for a "tool" like this? What does the tool do?
- "We take the prompt concept $\theta^*$ to be unique in this paper to simplify the notation of the prompt distribution $p_\text{prompt}$." I'm not sure why this would be just to "simplify the notation;" is it not in place to ensure that the test prompt does not occur in the pre-training set? Perhaps "unique" is unclear.
- "The target distribution differs from the pretraining distribution p on the distribution of $h_\text{start}^\text{test}$." And not $\theta^*$ as well?
- In Eq. (4), $p_\text{prompt}$ is overloaded---it represents the joint, whereas before, it represented only the conditional.

#### Minor, general comments:
- The paper should discuss work on overparameterization (e.g., Zhang et al. (2017)) and overtraining (e.g., Power et al. (2021)), which do not change the train error significantly but do affect downstream performance. These lines of work are relevant to the observation that "the inductive bias of scale may improve in-context learning beyond memorizing the training data better."

### References

Helske S, Helske J (2019). “Mixture Hidden Markov Models for Sequence Data: The seqHMM Package in R.” Journal of Statistical Software, 88(3), 1–32. doi:10.18637/jss.v088.i03

Power, Alethea, Yuri Burda, Harri Edwards, Igor Babuschkin, and Vedant Misra. "Grokking: Generalization Beyond Overfitting on Small Algorithmic Datasets." In ICLR MATH-AI Workshop. 2021.

Zhang, Chiyuan, Samy Bengio, Moritz Hardt, Benjamin Recht, and Oriol Vinyals. "Understanding deep learning requires rethinking generalization (2016)." arXiv preprint arXiv:1611.03530 (2017).

**Summary Of The Paper:**

The submission studies the phenomenon of the "in-context learning" behavior of large language models (LLMs). Two results are presented: First, the submission argues that in-context learning can be formalized as Bayesian inference in a mixture of Hidden Markov Models (HMMs). Second, the submission introduces small-scale settings in which the in-context learning behavior of LLMs is reproduced empirically.

**Summary Of The Review:**

The question (S1) is highly relevant, and I think the small-scale reproduction (S2) could be meaningful. However, the implications of what is shown in the paper for LLMs is not sufficiently clear (W5), so the significance of the work in answering the question (S1) is not clear. The paper also has a few other weaknesses that prevent me from recommending acceptance in its current form (W1-4).

### Update after discussion

Thanks to the authors for their detailed response. My concerns have been largely addressed and I appreciate the greater empirical connection to LLMs via the additional experiments; I have updated my score to reflect this. I do have recommendations to the authors to increase the clarity of the paper, which I do think should be improved for the next revision:

- I agree with Reviewer BHn6 that the title overclaims and does not reflect the contributions of the paper. The current version is suggestive of something like "An Explanation of In-context Learning [in Large Language Models like GPT-3] as Implicit Bayesian Inference," which is *not* the contribution of the paper, because the paper explains in-context learning in a simplified theoretical model, and provides some  empirical evidence that this captures important aspects of the same in LLMs, but does not strictly "explain" the same in LLMs.
- Related to the above, everywhere in the text where the term "language model" is used should have greater precision on the distinction between the idealized setting and the practical setting, because this term is used interchangeably for both the theoretical model and a large language model.
- I find the terms "in-context predictor" (theoretical notion) and "in-context learning" (behavioral notion) confusable. It would be worth writing "*ideal* in-context predictor" and "*behavior of* in-context learning" to make this less so.
-  OOD = low probability is an imprecise notion and one that does not seem to be connected to how the structure of the prompts is used in the analysis. I think the authors can improve on this point by clarifying what they mean by "we can no longer factorize the examples under the pretraining distribution."

---

> ### Author Response · Authors · 2021-11-18
> **Prompts are OOD, which is the basis for the novel theoretical results**
>
> We thank the reviewer for the detailed feedback. Reviewer j2L6 notes that the “paper studies a timely question” with a theoretical model that “may lead to new understanding”, and that the “existence proof of in-context learning in small-scale models is new and surprising”.
>
> Before we respond to the concerns, we give a high-level overview.
> - **Setup**: ​​We assume that pretraining data distribution is a mixture of HMMs, and the language model learns the pretraining distribution perfectly. Our main result is that we show that this language model can do in-context learning despite the prompts being out-of-distribution (OOD) with respect to the pretraining distribution.
> - **Framework**: At a high level, we prove that this in-context learning can emerge from modeling long-range coherence in the pretraining data. We assume all sentences in a pretraining document share a latent concept (long-range coherence). To generate coherent next tokens in LM pretraining, the LM must infer the latent concept. Inferring a shared concept from examples in a prompt leads to in-context learning.
> - **j2L6 was unclear about why the prompts are OOD**, which leads to the main concern about the significance/novelty of the theoretical results. At a high level, the prompts are OOD due to generating independent examples and concatenating them together, creating unnatural transitions between examples. We clarify this below.
>
> We address the concerns below:
>
> > j2L6 is “unsure about the significance/novelty of the theoretical results” and is “not sure they follow why... the test prompt is OoD” with respect to the pretraining distribution.
>
> We thank the reviewer for raising this important question. The prompts are OOD due to generating independent examples and concatenating them together, creating unnatural transitions between examples. The theory is in a novel setting since the prompts are OOD, and existing theory does not handle this. To clarify:
> - **Why is the prompt OOD?** We illustrate this with a simplified example. Suppose a pretraining document is “a b c d … x y z \n a b c d … y z a b c d ...” where the latent concept enforces the tokens to be in alphabetical order with high probability. A prompt is generated by first **independently** generating examples (e.g., “a b” , “x y” , “f g”) with random starting tokens drawn from a prompt start distribution. The prompt concatenates these examples, separated by a delimiter (“a b \n x y \n f g \n c” ) with a test input “c” at the end. The **transitions between examples can be unnatural** (e.g., “b” rarely transitions to delimiter, “x” rarely follows so closely after “b”). In general, there may not be a concept that places high probability on transitioning to a delimiter token every $k$ tokens. This means that **prompts are usually very low probability (OOD) sequences** under the pretraining distribution.
> - **Our theorems operate in a novel theoretical setting that hinges on the prompts being OOD**: Previous theory on Bayesian asymptotics (Bernstein von Mises-type results) either assume 1) independence of the observations or 2) a sequential observation drawn from the same HMM distribution. For 1): even though the examples were independently generated under the *prompt distribution*, we can no longer factorize the examples under the *pretraining distribution*, which does not have the factorized structure over examples. For 2): existing results do not extend to OOD sequences. **Thus the previous theory does not apply in our setting.** Theorem 1 deals with these challenges.
>
> >  “The theoretical model considered in the paper seems to be a mixture Hidden Markov Model (HMM). Helske & Helske (2019) demonstrate that such a model can be expressed as a standard HMM, for which inference is known to be possible and moreover efficient. Given this, it is not clear to me what Theorem 1 adds”
>
> We study HMMs not for any computational reason such as efficient inference, but just to make theoretical analysis feasible. Our **Theorem 1 does not describe how to do efficient inference** in the model (as in Helske & Helske), **but rather studies the asymptotic statistical behavior** of the pretrained LM when evaluated on prompts with an increasing number of examples.

---

> > ### Author Response · Authors · 2021-11-18
> > **Response to other Major comments (1/2)**
> >
> > > j2L6 would like clarification on which parts refer to the ideal setting with infinite pretraining data vs. empirical models and data.
> >
> > We thank the reviewer for the suggestion, and will clarify these differences in the paper. The setup of the paper is that we make an assumption about the data-generating process (mixture of HMM) and show how this data assumption can enable in-context learning. This data assumption holds throughout the paper. We assume that the pretrained LM fits the pretraining distribution exactly with enough pretraining data and expressivity. Thus **in the theory, the pretrained LM is the same as the pretraining distribution.** The main **role of the experiments is to relax this assumption** about the pretrained LM fitting the pretraining distribution exactly - we pretrain Transformers and LSTMs on data generated by the mixture of HMMs to approximate the pretraining distribution and see that the properties of the pretraining distribution transfer to these pretrained LMs.
> >
> > > j2L6 would like clarity on the term “in-context predictor”.
> >
> > We apologize for any confusion about this term. The “in-context predictor” precisely refers to the theoretical estimator that outputs the most likely next token given the prompt under the pretraining distribution. However, we do not mention the term “in-context predictor” outside of Section 2 and 3, which are about theory. When we describe the general behavior of learning a task from context, we call this “in-context learning”. However, if we missed something here, we’re happy to change it.
> >
> > > “If in-context learning just refers to improvement in the number of test examples (ie. the behavioral definition), then these results are quite distinct from the theoretical results. Can the asymptotic performance also be plotted to demonstrate that empirical performance approaches the ideal performance described in Section 3?”
> >
> > Thanks for the question. In-context learning does not only refer to improvement in the number of test examples, but also generally achieving learning/low error using examples given as a text input (which was also previously unclear). In our theory, we show that the asymptotic error goes to 0 (Theorem 1) or goes to zero with increasing example length $k$ (Theorem 2, 3). This implies that the error must go down with the number of examples, constituting in-context learning. From our theoretical results, we have a bound on the asymptotic accuracy, but these bounds are usually off by some constant factors, making the exact comparison between theory and experimental plots hard to do. Finite sample analyses are possible but will likely require different techniques or assumptions - we leave this to future work and view our current work as a first step.
> >
> > > j2L6 would like the paper to “more clearly discuss scope & limitations, and, in particular, which assumptions in the theoretical results and data model may not match reality”
> >
> > We thank the reviewer for this point. The high level scope is that **this paper focuses on the role of the pretraining distribution only** (since our assumptions are about the pretraining data and we assume the language models fit the pretraining distribution perfectly). Thus, some empirical phenomena about the role of model size and architecture are not covered with our framework.
> > - On particular assumptions, we believe that assuming infinite pretraining data and expressive enough models is reasonable, given the massive pretraining corpora that models like GPT3 are trained on and given the large size of GPT3 (175B parameters).
> > - On enumerable token sets, large models like GPT3 use a fixed vocabulary in practice so that this is a realistic setup.
> > - On the mixture of HMMs, this is a reasonably expressive but is indeed not the most realistic model of text - however, the important part is the presence of a latent variable which captures long-range (document-level) dependencies.

---

> > > ### Author Response · Authors · 2021-11-18
> > > **Response to Major comments (2/2) and Minor comments**
> > >
> > > > j2L6 comments that “​​the paper is a little light on take-aways for LLMs” (large language models)
> > >
> > >  We thank the reviewer for bringing this up and we will clarify the takeaways for LLMs in the paper. Many of the phenomena that we see (improved performance with number of examples, bigger models are better) are also seen in LLMs [1]. During rebuttals, we provide additional evidence (Section E.5, E.6) that GINC captures real-world in-context learning phenomena. **In Section E.5, we show that in-context accuracy in GINC is highly dependent on example ordering, similar to GPT-3** results from [2]. In Section E.6, we show that in some settings of GINC, **zero-shot can be better than one/few-shot accuracy but accuracy later improves with more examples, which mirrors the behavior of GPT-3** on some datasets (e.g., LAMBADA, HellaSwag, PhysicalQA, RACE-m) [1]. This is possible since one/few-shot prompts introduce the unnatural/OOD transitions between examples.
> > > - Conceptually, the **theoretical understanding helps with reasoning about the issues of prompt structure** (how it makes prompts OOD and how it affects in-context accuracy) as well as previously unexplored aspects such as the length of each example. In the appendix (Section E.7) we run an **experiment on GPT3 which studies the effect of example length** on the LAMBADA task. We find that longer examples improve in-context performance even when the test examples are short, which supports the theory.
> > > - Because we generate a **synthetic dataset, we can also control the latent properties of the data (e.g., out of distribution concepts) which can make predictions about LLMs** - we predict that completely unseen concepts cannot be in-context learned (but it's possible that the model can learn to combine some learned concepts together).
> > >
> > > [1] Tom Brown et al. Language models are few-shot learners. 2020
> > >
> > > [2] Tony Z. Zhao, Eric Wallace, Shi Feng, Dan Klein, Sameer Singh. Calibrate Before Use: Improving Few-Shot Performance of Language Models. ICML, 2021.
> > >
> > >
> > >
> > > Minor: We thank the reviewer for the detailed comments. We will improve these points in the paper, and also address some questions below.
> > > - We will elaborate in the paper on what we mean by "ideally, the posterior predictive distribution should concentrate". This is tied to the Bernstein-von Mises theorem, which is another point that the reviewer felt should be elaborated. The Bernstein-von Mises theorem would be the canonical method for analyzing the posterior distribution of a latent variable given observations, and then connecting the posterior distribution to frequentist notions (like saying that the posterior distribution concentrates to the frequentist MLE). We operate in a novel setting where Bernstein-von Mises doesn’t hold, but we want a similar type of result.
> > > - Confusion about “We take the prompt concept θ∗ to be unique in this paper”: we apologize for the confusion - we mean that the prompt distribution is defined with respect to some θ∗, and one could imagine considering many different prompt distributions with different prompt concepts θ∗. To simplify notation, we consider some particular prompt distribution and fix θ∗ (without loss of generality).
> > > - The reviewer asks, “ ‘The target distribution differs from the pretraining distribution p on the distribution of hstarttest.’ And not θ∗ as well?” The reviewer is correct that  θ∗ is also a difference and we’ll clarify this.
> > > - Overloading pprompt: In Eq. 2, we defined pprompt as generating the full joint and in Eq. 4 we use the conditional. We will clarify this.
> > > - We thank the reviewer for the reference suggestions and we will discuss the overparameterization work in the discussion section.

---

> ### Author Response · Authors · 2021-12-02
> **Further discussion**
>
> Dear reviewer, may we ask if you could respond to our comments? In our response below, we clarify why prompts are out-of-distribution, meaning that the theory operates in a novel setting. We also provide clarification and additional experiments on GINC and GPT-3 that strengthen the connections and takeaways for large language models. We also addressed other detailed concerns. Please let us know if you have other questions or concerns. Thank you!

---

> > ### Comment · Reviewer_j2L6 · 2021-12-06
> > **Thanks for your response**
> >
> > Thank you for the extensive response! I have updated my score / review; please see there.

---

### Author Response · Authors · 2021-11-18
**General response**

We thank all the reviewers for their thorough reviews. The reviewers thought the **“paper studies a timely question”**, the **“formal developments are extensive”** and **“really interesting to understand in what conditions few-shot prompting might work”**, the **“existence proof of in-context learning in small-scale models is new and surprising”** and presented as **“clear set of experiments”**.

Summary of common concerns:
- **Reviewers j2L6, BHn6 were unclear why prompts are out-of-distribution** with respect to the pretraining distribution and thus were concerned about the novelty/significance of the theoretical results. This is due to generating independent examples and concatenating them together, creating unnatural transitions between examples. We clarify this point in the individual comments.
- **Connections between our framework and GPT-3/large language models on real data”**. Our theory gives a framework for thinking about the critical components of in-context learning, and our small-scale GINC dataset which replicates the in-context learning phenomenon and the effects of model scaling was created using this understanding. To further strengthen the connections to large language models, we ran two additional experiments in the rebuttal (described in the changelog below), mirroring two phenomena from GPT-3 (sensitivity to example ordering and zero-shot better than one-shot in some instances) in our small-scale setting.

We addressed all the concerns in the individual comments. We list a summary of the changes we made to the paper here (major changes colored blue in the paper):
- **Strengthen connections to real data / large language models**: During rebuttals, we added two experiments where we see phenomena mirroring those seen in GPT3. In Section E.5, we show that in-context accuracy in GINC is highly dependent on example ordering, similar to GPT-3 results from Zhao et al 2021 [2]. In Section E.6, we show that in some settings of GINC, zero-shot can be better than one/few-shot accuracy but accuracy later improves with more examples, which mirrors the behavior of GPT-3 on some datasets (e.g., LAMBADA, HellaSwag, PhysicalQA, RACE-m) [1]. In the discussion section (Section 5), we detail some takeaways for in-context learning with real models and data.
- **Strengthen GPT3 experiment with additional baselines**: In Section E.7, we run an experiment on GPT3 which studies the effect of example length on the LAMBADA task. We find that longer examples improve in-context performance even when the test examples are short, which supports the theory. We add baselines where we double the number of short examples to equalize the total prompt length with the longer examples.
- **More detailed description of GINC**: We add a more detailed description of the generating process of GINC in Section 4 and in Appendix E.1. We also add a concrete example of pretraining data and prompt from GINC in Figure 7. We also discuss the relation between GINC and the theoretical assumptions.
- **Improve explanation of why prompts are OOD**: We improve the introduction, Section 2 text to make it clearer why the prompts are out of distribution with respect to the pretraining distribution.
- **Simplified Figure 1**: We changed Figure 1 to describe the in-context learning framework we consider at a higher level, which we hope will be more informative for the reader in the beginning of the paper.


[1] Tom Brown et al. Language models are few-shot learners. 2020

[2] Tony Z. Zhao, Eric Wallace, Shi Feng, Dan Klein, Sameer Singh. Calibrate Before Use: Improving Few-Shot Performance of Language Models. ICML, 2021.

---

### Decision · Program_Chairs · 2022-01-20

**Decision:**

Accept (Poster)

**Comment:**

This submission introduces a theoretical model to explain how "in-context learning" (i.e. the ability to output a correct prediction based on inputs for a task that the model was not explicitly trained on) is possible. The model uses a mixture of HMMs and shows that in-context learning is a natural consequence of Bayesian inference under that model. Overall, reviewers agreed that the contribution was useful and timely, and were somewhat convinced by the theoretical arguments. However, there was some broad concern with the framing of the paper. Namely,
1) The paper claims that prompted data is OOD w.r.t. the pre-training distribution. In fact, this is almost certainly not the case for many tasks and datasets. Indeed, it is highly plausible that data very similar to the example given by the paper (identifying the nationality of different celebrities) appears in the pre-training dataset of large LMs. Other examples include the popular "tldr;" task format for summarization which is incredibly common on the internet, etc.
2) The paper does not sufficiently distinguish between insights gained in the toy setting considered by the theoretical model and insights that can be applied to large LMs. Most reviewers were concerned that there might not be any reason to think that the insights gained from the theoretical model would apply to large LMs. The paper, however, very much frames itself as developing insight into the behavior of large LMs.

I will recommend acceptance of this paper, but will stipulate that the above two issues should be fixed in the camera-ready version. Namely, I would suggest that the authors do not refer to prompted forms of tasks/datasets as "OOD", and I would suggest that any claims about different insights are not applied to large LMs.